# Parallel development of social behavior in biological and artificial fish

Joshua D. McGraw [1,2,4] ✉, Donsuk Lee [1,4] & Justin N. Wood [1,2,3] ✉

Our algorithmic understanding of vision has been revolutionized by a reverse engineering paradigm that involves building artificial systems that perform the same tasks as biological systems. Here, we extend this paradigm to social behavior. We embodied artificial neural networks in artificial fish and raised the artificial fish in virtual fish tanks that mimicked the rearing conditions of biological fish. When artificial fish had deep reinforcement learning and curiosity-derived rewards, they spontaneously developed fish-like social behaviors, including collective behavior and social preferences (favoring in-group over out-group members). The artificial fish also developed social behavior in naturalistic ocean worlds, showing that these embodied models generalize to real-world learning contexts. Thus, animal-like social behaviors can develop from generic learning algorithms (reinforcement learning and intrinsic motivation). Our study provides a foundation for reverse-engineering the development of social behavior using image-computable models from artificial intelligence, bridging the divide between high-dimensional sensory inputs and collective action.

A core scientific goal is to understand the algorithms underlying biological intelligence. Recently, a new reverse engineering paradigm has transformed our algorithmic understanding of perception (e.g., vision[1], audition[2], and olfaction[3]), action (e.g., walking[4], soaring[5], and visually-guided grasping[6]), and higher-level cognitive abilities (e.g., language[7], navigation[8,9], and memory[10]). These success stories were fueled by breakthroughs in artificial intelligence, where it is now possible to build artificial neural networks (ANNs) that perform the same tasks as humans, in some cases achieving human levels of performance. Since ANNs perform the same tasks as humans, and use computational machinery that is based on neuron-like units, ANNs can reveal which learning algorithms are necessary and sufficient to solve psychological tasks. ANNs can be directly compared to biological systems (e.g., on an image-by-image basis), allowing ANNs to serve as runnable, neurally mechanistic, and computationally explicit hypotheses of the algorithms underlying animal intelligence[11].

Despite these strengths, there are significant differences between ANNs and animals that prevent direct comparisons between artificial and biological systems. The ANNs typically used to reverse engineer perceptual and cognitive abilities are disembodied passive consumers of data, waiting to be spoon-fed the right sorts of experiences for learning. Most ANNs also do not have bodies, and their behavior typically consists of applying a limited stock of labels to static images (e.g., object recognition tasks) or strings of text (i.e., NLP tasks). Conversely, animals have bodies with rich behavioral repertoires. Animals choose where to go and what to look at, producing developmentally-structured datasets for optimizing learning across their lifespan[12]. Biological intelligence also emerges over multiple, nested timescales as animals learn to ground knowledge in real-world experience by interacting with the environment in purposeful ways[13].

Recent studies have addressed some of these concerns. In vision, for example, researchers have shown that self-supervised algorithms learn core object recognition skills when trained 'through the eyes' of human infants[14–16] and newborn chicks[17,18]. These findings suggest that disembodied ANNs can learn animal-like abilities when trained with biologically plausible data (embodied data streams), in the absence of

[1]Department of Informatics, Indiana University Bloomington, Bloomington, IN, USA. [2]Cognitive Science Program, Indiana University Bloomington, Bloomington, IN, USA. [3]Department of Neuroscience, Indiana University Bloomington, Bloomington, IN, USA. [4]These authors contributed equally: Joshua D. McGraw, Donsuk Lee. ✉e-mail: jdmcgraw@iu.edu; woodjn@iu.edu

labels and supervision. Researchers have also built embodied computational models to explore how ANNs learn in more realistic environments[19–25]. These studies suggest that some neural findings (e.g., head direction cells, grid cells, place cells) and behavioral skills (e.g., path integration, ego motion, object-based attention) emerge spontaneously when ANNs are trained in embodied learning contexts.

To date, however, it is unclear whether ANNs are accurate models in the embodied learning conditions faced by newborns. ANNs are widely assumed to be data hungry, requiring vast amounts of training data to develop brain-like intelligence[26–28]. Conversely, newborn animals rapidly learn to solve challenging tasks, with many abilities emerging within the first few days[29–32]. Controlled-rearing studies, which involve controlling the experiences (training data) available to newborn animals, provide particularly striking examples of the power and efficiency of newborn brains. For instance, newborn chicks rapidly develop high-level vision, including object segmentation[30], object recognition[31], and object permanence[32]. Newborn animals also rapidly develop social knowledge, including face recognition[33] and selective attention towards social partners[34,35]. Can embodied ANNs account for the rapid feats of learning observed in newborn animals?

To address this question, we argue that the field needs 'embodied developmental benchmarks' focused on reverse engineering the core learning algorithms that power biological intelligence. These benchmarks must have two features. First, the animals and ANNs must be raised in the same environments. Intelligence emerges both from the learning algorithms and training data acquired by agents. Thus, differences in intelligence across animals and ANNs could be due to differences in the learning algorithms or training data. Evaluating whether ANNs learn like animals requires 'raising' embodied ANNs in

the same environments as animals and giving ANNs and animals access to the same experiences. Second, the animals and ANNs must be tested with the same tasks. Psychologists have long recognized that measures of intelligence are task-dependent[36–38]. Confirming that animals and ANNs learn the same abilities requires testing them with the same tasks, ensuring that differences across the animals and ANNs were not due to differences in the tasks themselves.

Here, we describe digital twin experiments that meet both requirements, allowing newborn animals and ANNs to be raised in the same environments and tested with the same tasks (Fig. 1). Digital twin experiments involve first selecting a target animal study, then creating digital twins (virtual replicas) of the animal environments in a video game engine. Artificial animals (embodied ANNs) are then raised and tested in those virtual environments and their behavior is compared to the behavior of the biological animals in the target study. By raising and testing animals and ANNs in the same environments, we can measure whether animals and ANNs spontaneously develop along common pathways and produce common learning outcomes.

We focused on social behavior. Animals are inherently social and organize spontaneously into cohesive groups, such as insect swarms[39], bird flocks[40], fish schools[41], and human crowds[42]. Many animals begin grouping early in life, within days to weeks after birth[43].

To study collective behavior, researchers typically use rule-based models, where individuals are modeled as featureless points that change their behavior according to a fixed set of hard-coded interaction rules[44,45]. Rule-based models have provided many valuable insights into collective behavior[46–49]. However, rule-based models are not well-suited for studying the learning algorithms and motivational systems that drive collective behavior because these models typically

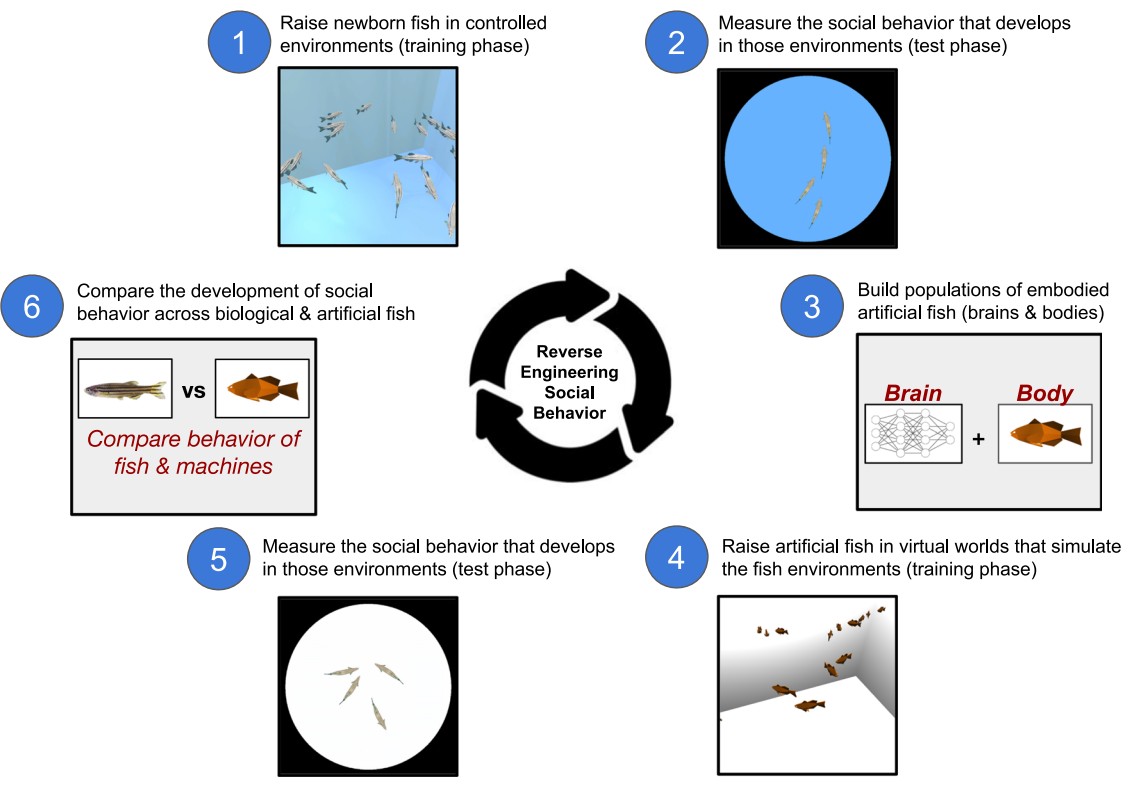

**Fig. 1 | Digital twin method for comparing the development of social behavior across biological and artificial fish.** First, biological fish were reared and tested in controlled environments (Steps 1–2). Second, we built populations of artificial fish (Step 3), then reared and tested them in digital twins (virtual worlds) that simulated the fish environments (Steps 4–5). Third, we compared the social behavior of the biological and artificial fish (Step 6). We used this method to evaluate whether artificial fish learn fish-like collective behavior and social preferences when raised in the same visual environments as biological fish. We use a circular diagram to emphasize that pixels-to-actions models allow scientists to build models that learn like newborn animals in one study, then iteratively add new studies to the testbed to validate and refine those models.

lack learning algorithms and motivational systems. Rule-based models also do not typically operate over raw sensory inputs; rather, they operate over high-level features (e.g., position and orientation of neighbors) that are not directly available from sensory signals. The models thus make simplifying assumptions about the nature of visual learning and behavior. To develop collective behavior in the real world, animals must learn to convert high-dimensional sensory inputs (~$10^6$ optic nerve fibers) into a manageably small number of perceptually relevant features to drive adaptive action.

To address these limitations, we built embodied pixels-to-actions models of collective behavior that learn from raw sensory inputs, make decisions, and perform actions, driven by self-supervised learning objectives. We then compared the development of collective behavior across these pixels-to-actions models and newborn fish. We chose fish because they can be reared in controlled environments, are mobile on the first day after hatching, and rapidly learn collective behavior based on visual information[43,50–52].

The biological fish and computational models (artificial fish) shared three constraints: (1) both were embodied and performed actions in 3D environments; (2) both learned from raw visual inputs; and (3) both learned through self-supervised objectives. This placed strong constraints on learning, forcing the ANNs in the artificial fish to solve similar problems as the brains in biological fish. Both biological and artificial fish needed to learn to convert high-dimensional sensory signals into adaptive actions in complex visual and social environments, guided solely by intrinsic motivation.

Here, we show that these constraints are sufficient to produce fish-like social behaviors in embodied ANNs. We support this conclusion in experiments on collective behavior (Experiments 1-2) and social preferences (Experiment 3).

## Results
### Experiment 1: Collective behavior
For the target animal study, we chose Hinz & de Polavieja[43], who discovered that collective behavior develops gradually in zebrafish. Fish began turning toward each other around 7 days post-fertilization (dpf) and increased the intensity of social interactions until 3 weeks of age. Hinz & de Polavieja[43] explained this pattern with a simple attraction rule, in which fish attract each other part of the time, with attraction defined as the behavior of turning toward another randomly chosen fish. Over time, fish spend more time in attraction behavior, thereby increasing attraction strength across development. We explored whether ANNs move along a common developmental trajectory as zebrafish when they are embodied in artificial fish and raised in virtual fish tanks.

We modeled the virtual fish tank after the fish tank described in the target animal study[43] (Methods). We built the artificial fish by embodying self-supervised learning algorithms in artificial fish (Methods). Each fish received raw visual input through an invisible forward-facing camera attached to its head. The artificial fish could move around the virtual tank by selecting an action on every time step (Methods). Movement was restricted to a single flat plane in order to mimic the action space of a thin, shallow, layer of water commonly used in zebrafish research to prevent motion along the height axis.

The artificial fish had two biologically inspired learning algorithms: (1) reinforcement learning (RL) and (2) intrinsic motivation. In RL, agents maximize their long-term rewards by performing actions in response to their environment and internal state. To succeed in environments approaching real-world complexity, agents must learn abstract and generalizable features to represent their environment. To this end, deep RL combines RL with deep neural networks to transform raw sensory inputs into efficient representations capable of supporting adaptive behavior[53].

The second algorithm—intrinsic motivation—provides a self-supervised reward landscape for optimizing deep RL. Prior studies

show that intrinsic motivation can drive the development of complex behaviors[24,54]. For example, one form of intrinsic motivation, curiosity-driven learning, promotes learning by motivating individuals to seek out informative experiences[55]. By seeking less predictable experiences, agents can gradually expand their knowledge of the world, continuously acquiring useful training data for learning[56,57].

To explore whether the development of collective behavior requires a particular type of intrinsic motivation, we tested intrinsic motivational algorithms spanning major classes of self-supervised learning used for deep RL (Methods). We used an Intrinsic Curiosity Module (ICM)[56], ICM with random features for encoding observations[57], Random Network Distillation (RND)[58], and Curious Representation Learning (CRL)[59] adapted from a popular contrastive learning algorithm[60,61]. Each algorithm takes batches of inputs and produces sets of rewards. The batches and rewards are used to train a policy network, which controls the moment-to-moment behavior of the artificial fish. For all algorithms, we used the same policy optimization algorithm: Proximal Policy Optimization (PPO)[62]. The policy network was optimized to maximize the reward generated by the intrinsic motivation algorithm.

Intrinsic motivation drives learning in humans and animals[63], including fish[64], so we hypothesized that intrinsic motivation and RL would drive the development of social behavior in artificial fish. Social partners are typically the least predictable things in a newborn's visual environment, so embodied agents equipped solely with intrinsic motivation and generic learning algorithms (e.g., deep RL) should learn to track and follow social partners. If our hypothesis is accurate, then artificial fish should develop common social behaviors as biological fish when artificial fish are equipped with ANNs that learn solely through deep RL and intrinsic motivation.

During training, we reared the artificial fish in virtual fish tanks and saved their ANN weights across 20 equally-spaced checkpoints. The 20 checkpoints corresponded to the 20 days of experimental data collected in the fish study[43]. We then compared each real day of development (biological fish) to each 'artificial day' (checkpoint) from the digital twin experiments.

To directly compare biological and artificial fish, our analyses parallel those used in the target animal study[43]. We first performed trials with pairs of fish, using a measure that involved calculating the distance between the two fish across time (Fig. 2a). Hinz & de Polavieja[43] reported that the distance between two fish gradually decreases as fish mature over the first 24 days. Likewise, the distance between the artificial fish decreased across the training phase, mimicking the developmental trajectory of biological fish (Fig. 2b). For all intrinsic motivation algorithms, the mean distance between the artificial fish was significantly lower than control randomized data (Fig. S1 and Methods). Supplementary Movie 1 shows the learned behavior across training. At first, the artificial fish behaved randomly, but by the end of training, they showed clear signs of collective behavior, spontaneously forming cohesive groups.

Since interactions depend not only on the distance between fish but also on their relative positions in space, we also measured the relative positions of the artificial fish using a coordinate system with its origin on a focal fish and the positive y axis pointing in the direction of its velocity vector. We then computed the probability of finding the second fish in space. As reported by Hinz & de Polavieja[43], the biological fish spent significantly more time with other fish in close side-by-side and front/back positions, and as the fish matured, they spent increasingly more time close to each other (Fig. S2a). Likewise, the artificial fish spent significantly more time with other fish in close proximity (typically front/back positions), and as the artificial fish learned during training, they spent increasingly more time close to each other (Fig. S2b). We observed the same pattern across all intrinsic motivation algorithms.

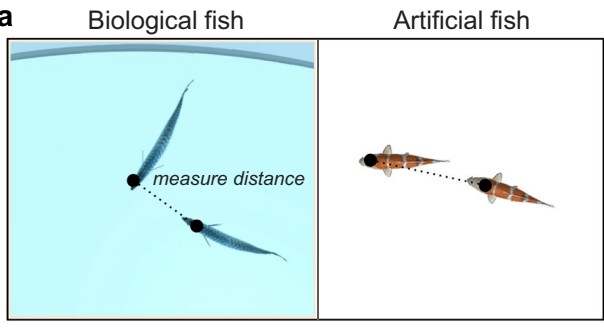

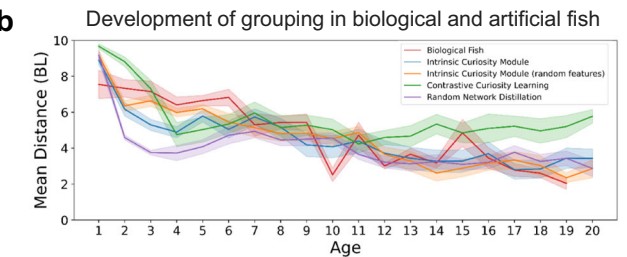

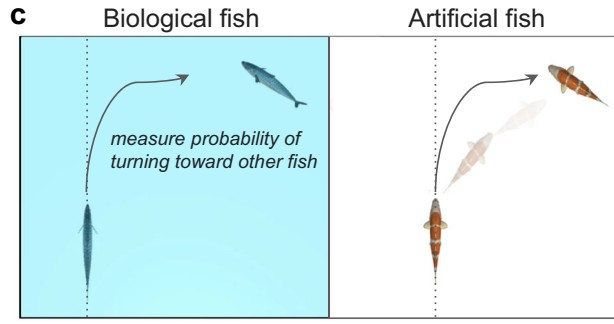

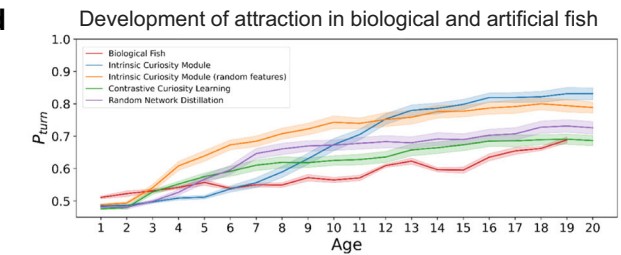

**Fig. 2 | Experiment 1 Results.** For each intrinsic reward algorithm, we trained artificial fish ($n = 20$) together in a virtual fish tank and saved the weights in their neural networks at 20 equally spaced time points during training. "Age" is defined as the index of the checkpoint in chronological order. **a** To measure whether the artificial fish developed fish-like grouping behavior, we measured the distance between pairs of biological fish and artificial fish across the training phase. **b** Both biological and artificial fish developed common grouping behavior across time. BL body lengths. **c** We also measured social attraction by studying the probability of turning toward the other fish (fish were tested in pairs). **d** Development of attraction: Probability of biological fish and artificial fish accelerating toward the side where another fish is located as a function of age. Data are presented as mean values. Error bars indicate ±1 SEM.

We did observe a difference between the biological and artificial fish in terms of the relative placement of other fish: the biological fish spent more time with other fish in both front/back and side-by-side alignments, whereas the artificial fish tended to spend more time with other fish in a front/back alignment. This likely occurred because the fish had two eyes, one on each side of the body, while our artificial fish had one eye, positioned at the front of the body. The biological and artificial fish may have followed the same strategy (i.e., "keep social partners in view"), leading biological fish to position social partners to their side and artificial fish to position social partners in front.

Next, we measured whether biological and artificial fish repel and attract social partners in similar ways[65,66]. Following Hinz & de Polavieja[43], we computed the probability that the focal fish turned to the side with the other fish (Fig. 2c). Biological fish begin turning toward another fish early in development (7 days) and the probability of this behavior becomes stronger across development. The artificial fish showed a similar pattern, with attraction toward other fish developing gradually across training (Fig. 2d). We also studied social attraction by measuring the probability of turning to the right side depending on the specific position of the second fish in space (Fig. S3a). The biological (Fig. S3b) and artificial (Fig. S3c) fish developed common sensitivities to the relative position of other fish in space.

We then measured collective behavior with four fish (Fig. 3a). Hinz & de Polavieja[43] found that their data could be explained by a model in which each fish interacts by moving towards a randomly chosen fish during certain periods of time. This model states that when a focal animal has $N_1$ animals to one side and $N_2$ animals on the other side, the probability of choosing the side with $N_1$ animals is

$$P(N_1 | N_1 : N_2) = p_s \frac{N_1}{N_1 + N_2} + (1 - p_s)\frac{1}{2} \quad (1)$$

and the probability of choosing the other side is $P(N_2 | N_1 : N_2) = 1 - P(N_1 | N_1 : N_2)$.

This model makes specific predictions for groups of four fish, in which a focal fish can be found having no fish on one side and three fish on the other side (configuration 0:3), or one and two fish to either side (configuration 1:2). The model predicts a relationship for 0:3 and 1:2 configurations that is independent of the one parameter in the model ($p_s$ = time spent in interactions):

$$P(1 | 1 : 2) = \frac{1}{3} + \frac{1}{3}P(0 | 0 : 3) \quad (2)$$

plotted as a dashed line in Fig. 3b. Both the biological and artificial fish developed social attraction behavior that accorded with the theoretical predictions in Eq. (2). We observed the same pattern in all intrinsic motivation algorithms.

We also tested whether the predictions of this model generalize to larger groups (Fig. 3c). The interaction rule in Eq. (1) predicts that the probability of turning to the side with $N_1$ fish grows linearly with $N_1$ as $N_1 / (N_1 + N_2)$. Following Hinz & de Polavieja[43], we tested artificial fish in groups of seven, since there are four configurations arising in groups of seven fish (3:3, 4:2, 5:1, 6:0). Like biological fish, development in the artificial fish largely accorded with predictions of the interaction rule (Fig. 3d).

Finally, we tested whether the trained artificial fish behaved like adult fish (tested 150 days post-fertilization[43]). Following Hinz & de Polavieja[43], for each day $t_i$, we computed three values for $p_s(t_i)$ by fitting Eq. (1) to the experimental probabilities $P(2 | 2:1)$, $P(0 | 0:3)$, and $P(0 | 0:1)$ of that day and computing its mean value (Fig. 4a). In biological fish, the attraction parameter $p_s$ increased from 0.01 (6 days) to 0.47 (24 days), and then increased to 0.54 by the adult stage (red dashed line in Fig. 4b). Likewise, the artificial fish showed an increase in this attraction parameter in the early and middle stages of training, before the attraction parameter reached values similar to (or stronger) than the values observed in adult fish. Thus, biological and artificial fish show common increases in the amount of time spent in social interaction. By the end of learning/training, both biological and artificial fish spontaneously developed robust collective behavior.

There was one difference between the artificial fish and the biological fish in the target animal study. Hinz & de Polavieja[43] reported

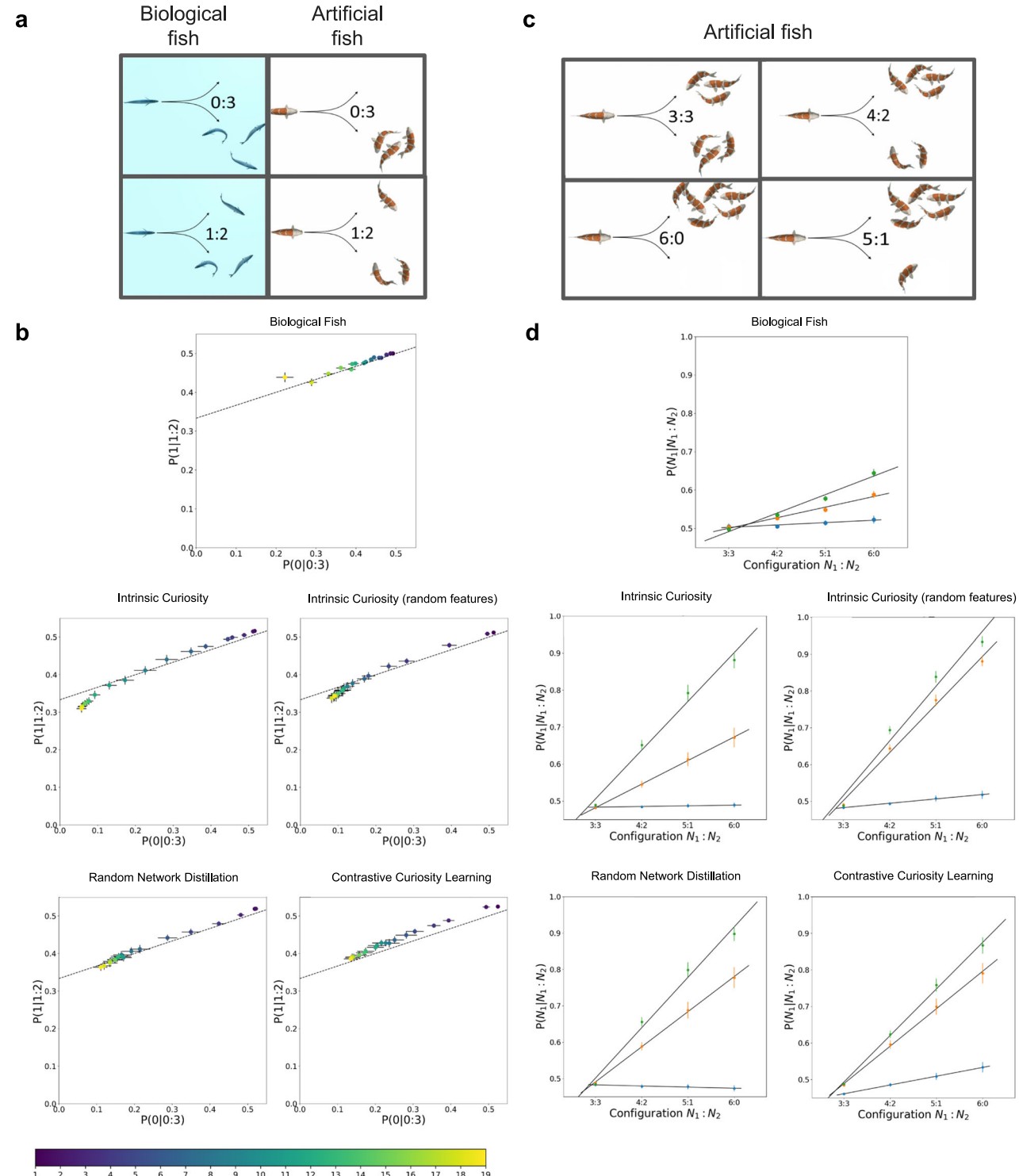

**Fig. 3 | Experiment 1 Results. a** We studied collective behavior in groups of four fish, measuring the probability of turning left versus right depending on the number of fish on each side. **b** Common attraction rules in biological fish (top) and artificial fish (bottom 4 graphs). Relationship between the probability of turning to the side with 1 fish in a configuration (1:2), $P(1 \mid 1{:}2)$, and the probability of turning to the side with no fish in a configuration (0:3), $P(0 \mid 0{:}3)$. Dots indicate experimental data (color coded by age). The dashed line indicates the theoretical line derived

from the interaction rule in Hinz & de Polavieja[43]. **c** We also studied collective behavior in groups of seven fish, measuring the probability of turning left versus right depending on the number of fish on each side. **d** Probability of turning to the side with three, four, five, and six biological fish (top) or artificial fish (bottom 4 graphs) in a group of seven fish, for ages 7 (blue dots), 11 (orange dots), and 15 (green dots). Data from the biological fish reproduced from Hinz & de Polavieja[43].

that biological fish were equally likely to turn toward other fish when there were three fish on one side and zero fish on the other side compared to when there was just one fish on one side and zero fish on the other side (Fig. S4). This pattern supports a simple attraction rule in

which fish turn toward another fish chosen at random. Conversely, when there were zero fish on the opposing side, we found that artificial fish were more likely to turn toward three fish than one fish (Fig. S4). This pattern indicates that the artificial fish followed a somewhat

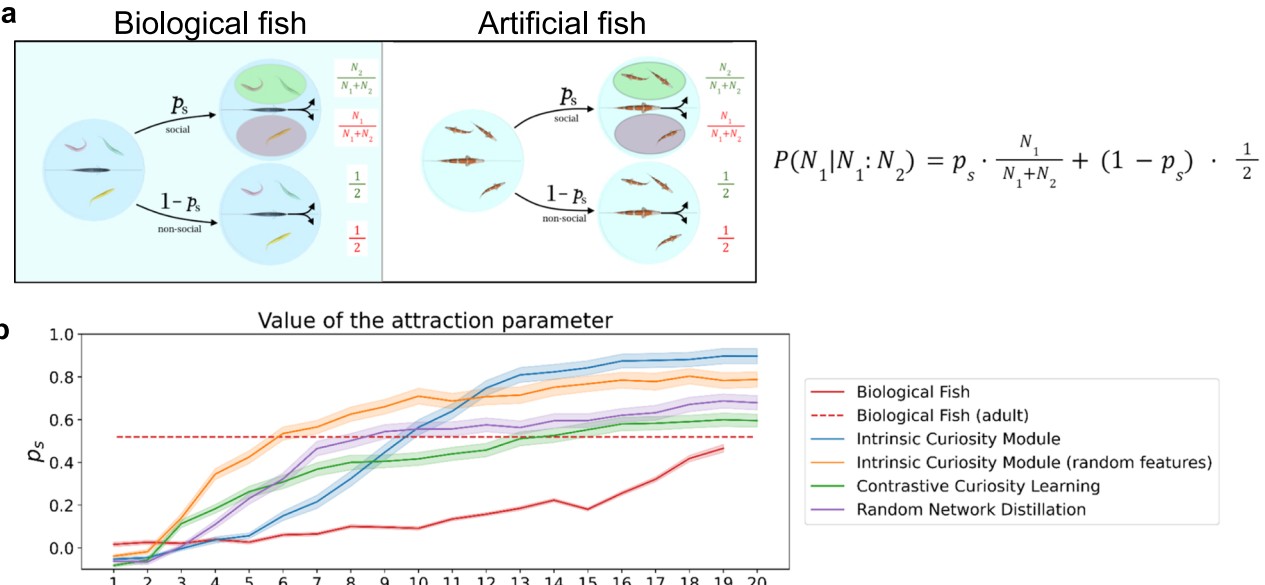

**Fig. 4 | Experiment 1 Results. a** Following Hinz & de Polavieja[43], for each day $t_i$, we computed three values for $p_s(t_i)$ by fitting Eq. (1) (right) to the experimental probabilities $P(2|2:1)$, $P(0|0:3)$, and $P(0|0:1)$ of that day and computing its mean value. **b** For biological fish (red lines) and artificial fish, the mean value of the attraction parameter ($P_s$ from Eq. (1)) increases during development. Most of the change in behavior can be explained simply by an increase in the amount of time spent in interaction across development. Data are presented as mean values. Error bars indicate ±1 SEM.

different attraction rule. Specifically, this behavior can be explained through the attraction rule proposed by Perez-Escudero & de Polavieja[67] (i.e., the probability given by their equation 17) to explain the shoaling behavior of a different fish species, three-spined sticklebacks (*Gasterosteus aculeatus*).

How did the development of collective behavior vary across the intrinsic motivation algorithms? We observed two patterns. First, all four intrinsic motivation algorithms were sufficient to learn collective behavior. This implies that the development of collective behavior does not require a particular type of intrinsic motivation. Collective behavior is an emergent property of a large class of intrinsic motivation algorithms.

Second, some intrinsic motivation algorithms developed stronger collective behavior than others (Fig. 4b). Despite these differences, all algorithms generated $P_s$ values (i.e., probability of engaging in social interactions) that were at least as strong as the adult biological fish[43] ($P_s = 0.54$). These intrinsic motivation algorithms were thus sufficient to mimic the rapid development of collective behavior observed in biological fish.

## Experiment 2: Collective behavior in naturalistic worlds
To confirm that artificial fish can learn collective behavior in more realistic visual environments akin to those faced by fish in nature, we tested whether the artificial fish learn to group when reared in naturalistic ocean worlds (Fig. 5a). We created a virtual seafloor world with high-resolution sand textures, shadows, drifting ocean particles, and caustic lighting. We also tested whether collective behavior develops in both blue and orange fish to confirm that the models can generalize across different fish pigments. Experiment 2 was not a digital twin experiment of a prior animal study, but rather a validity check that our embodied models can generalize to naturalistic learning contexts.

During training, we measured the average pairwise distance across all of the fish in the group. If the artificial fish developed collective behavior, then the distance between fish should have declined across the training period.

All four intrinsic motivation algorithms, and both blue and orange fish, rapidly developed collective behavior (Fig. 5b). The distance between artificial fish declined across training, confirming that these artificial fish can learn collective behavior in naturalistic visual worlds.

## Experiment 3: Social preferences
Unified scientific models should explain findings across many studies. This is one reason why pixels-to-actions models are valuable from a scientific perspective. Pixels-to-actions models can be reared and tested in the same environments as animals, so they can be directly compared to animals across a wide range of studies. We can select other studies from the literature, run digital twin experiments, then test whether the same model learns the same behaviors as animals across studies. This idea embraces the integrative benchmarking approach used to reverse engineer the ventral visual stream[11], but expands the approach to embodied learning contexts.

In Experiment 3, we selected a second controlled-rearing study with newborn fish that explored the development of social preferences[50]. We focused on social preferences for two reasons. First, collective behavior and social preferences are typically studied separately. For instance, the rule-based models used to study collective behavior are not commonly used to study social preferences, since rule-based models have hardcoded interaction rules, whereas social preferences are widely thought to be learned. Studying collective behavior and social preferences with the same pixels-to-actions model thus provides an opportunity to build unified models across fields at an engineering-level of abstraction (the level at which we build models that learn and behave like animals).

Second, social preferences are widespread in nature. Many animals, including humans, develop social preferences early in life, rapidly learning to favor "us" over "them" during social interaction[50,68–70]. Despite massive interest in social preferences across psychology and neuroscience, we know little about the core learning algorithms that generate social preferences in newborn organisms. What learning algorithms cause social preferences to develop so rapidly and flexibly early in life?

To tackle this question, we tested whether the core learning algorithms from Experiments 1-2 are sufficient to develop fish-like social preferences[43,50]. If so, then artificial fish should spontaneously

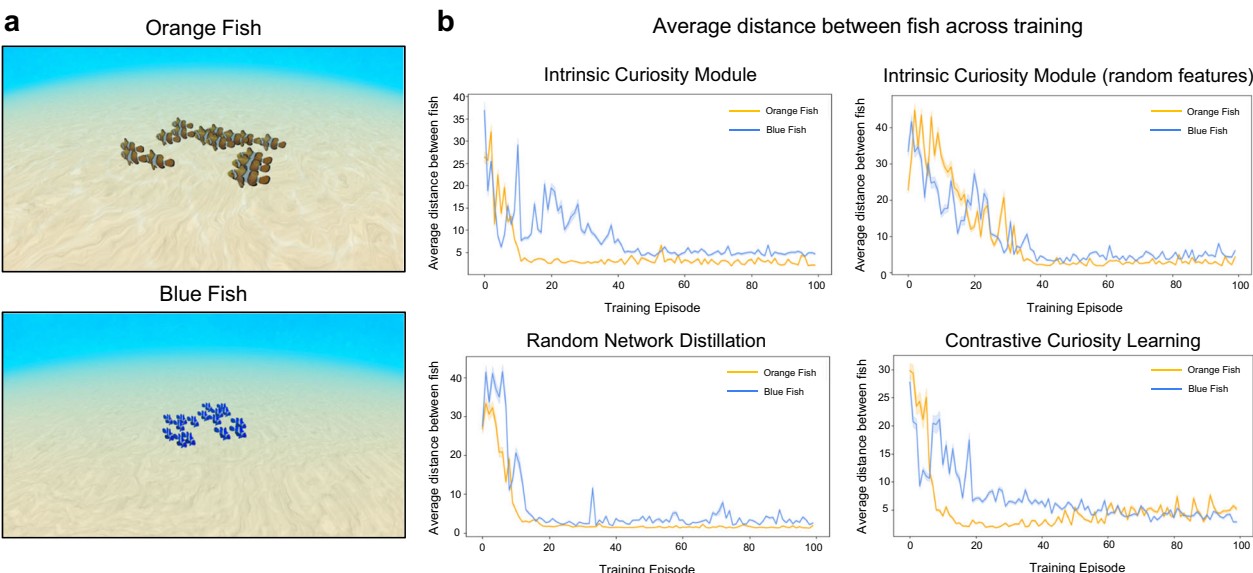

**Fig. 5 | Experiment 2 Methods and Results. a** We tested whether artificial fish can learn collective behavior when trained in naturalistic visual environments, akin to those faced by fish in nature. We created a realistic underwater seafloor environment with high-resolution sand textures, shadows, drifting ocean particles, and caustic lighting. **b** During the training period, we measured the average pairwise distance across all of the fish ($n = 16$) in the group. All of the artificial fish (all four intrinsic motivation algorithms and both the blue and orange fish), rapidly developed collective behavior early in training. Data are presented as mean values. Error bars indicate ±1 SEM.

develop "us vs. them" behavior, learning to prefer members of their group over members of other groups.

For the target animal study, we chose Engeszner, Ryan, & Parichy[50], in which newly-hatched zebrafish were reared (for several months) with fish of one of two pigments. Once the fish had been reared with social partners of one pigment, the researchers used a two-alternative forced-choice (2AFC) task to test whether the fish developed a preference for the familiar versus novel pigment. The fish developed a preference for familiar pigmented fish, independent of the fishes' own pigment. This experiment thus reveals a central role of learning in the development of social preferences.

To explore whether artificial fish develop fish-like social preferences, we performed a digital twin experiment, rearing and testing artificial fish in virtual replicas of the rearing conditions used for the biological fish[50] (Fig. 6 and Methods). To simulate the rearing conditions of the fish, we created two groups of differently colored artificial fish (blue and orange fish from Experiment 2), then reared four orange fish together in one group and four blue fish together in another group. Each group was reared in a white virtual cup, akin to the rearing conditions from Engeszer et al.[50] (Fig. 6a).

After training, we froze the ANN weights to mimic sensitive/critical periods observed in animals. Many animals have sensitive/critical periods which slow/stop learning, meaning part of an animal's behavior can be based on experiences that happened early in life, rather than recently. ANNs do not have sensitive/critical periods, which poses a problem for comparing animals to ANNs. One solution is to gradually slow the learning rate of ANNs across the training phase and freeze the weights when learning hits zero. We see this as roughly analogous to critical periods in brains, where learning can gradually slow and cease in animals.

We tested the artificial fish in two tasks. First, to mimic the testing conditions of the biological fish, we used the 2AFC task (Fig. 6a). On each trial, a single test fish was placed in the center of the chamber at a random orientation. We then measured whether the fish spent more time near a shoal with familiar pigmented fish versus a shoal with novel pigmented fish.

The artificial fish spent significantly more time near the familiar versus novel pigmented fish (Fig. 6b). We observed this pattern for all intrinsic motivation algorithms. In fact, the artificial fish developed stronger social preferences than biological fish. The fish in Engeszer et al.[50] spent ~62% of their time with in-group members, whereas most artificial fish spent almost all of their time with in-group members. The artificial fish also developed different levels of social preference, with some strongly preferring in-group members and others developing weaker in-group preferences. Despite being trained in identical environments, the artificial fish developed different social personalities.

In the real-world, social preferences drive animals to self-segregate into groups. To test whether our artificial fish self-segregate into groups, we created a second self-segregation task that involved placing all of the artificial fish in the same environment and measuring whether the fish spend more time with in-group versus out-group members (Fig. 7a). The self-segregation task was not a digital twin experiment of a prior animal study, but a validity check that the 2AFC task captures the construct under investigation: social preferences.

To test whether the fish self-segregated, we measured the distance between each fish and every other fish at each time step, then computed the in-group distance (average distance to familiar colored fish) and out-group distance (average distance to novel colored fish). The artificial fish spent significantly more time near fish with familiar versus novel colors (Fig. 7b), indicating that the artificial fish spontaneously learned to self-segregate into social groups.

To test whether intrinsic motivation was important for the development of social preferences, we created a new batch of artificial fish but reduced the strength of their curiosity reward from 1.0 to .001. We made no other changes. In these curiosity-reduced fish, we found weaker evidence for social preferences in the 2AFC task (Fig. S5a) and self-segregation task (Fig. S5b). Without a robust reward landscape to drive learning, the artificial fish generally did not develop social preferences.

Like Engeszer et al.[50], we also examined the importance of social experiences on the development of social preferences (Fig. 6c). Rather than rearing the artificial fish together during training, we reared them separately. These artificial fish acquired visual experience with the environment but did not acquire experience with other artificial fish.

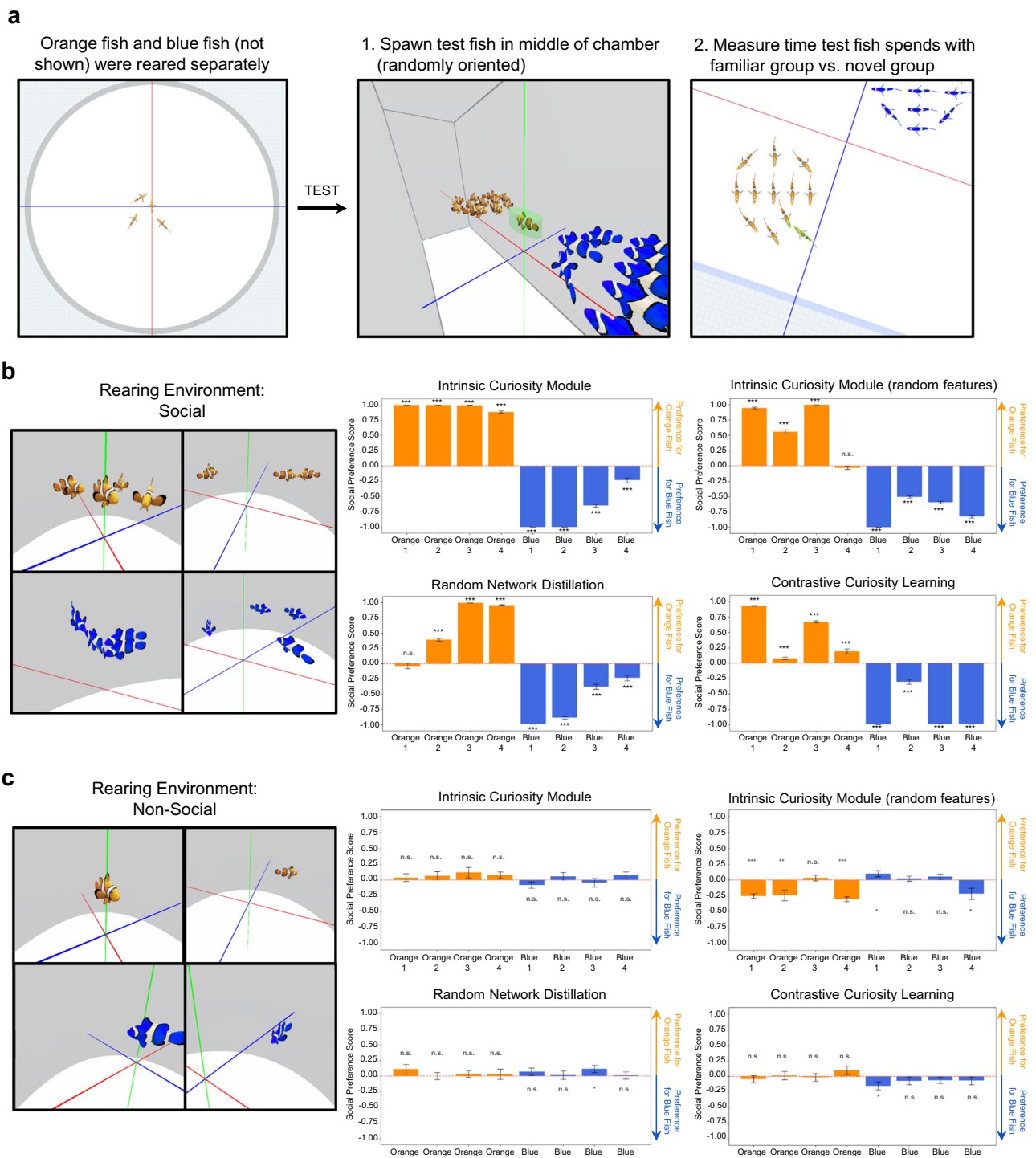

**Fig. 6 | Experiment 3 Methods and Results. a** We reared two groups (orange and blue fish) separately in small white virtual cups, akin to the rearing conditions from the biological fish study (Engeszer et al.[50]). After training, we froze learning in the artificial fish and tested their social preferences with the 2AFC task used in Engeszer et al.[50], measuring whether each fish spent more time with the familiar versus novel group. The red, green, and blue coordinate lines are for visualization and were not visible to the artificial fish. **b** Results from the 2AFC task. Each bar shows the social preference score (proportion of time spent with familiar versus novel group) of a single fish ($n = 8$). The fish reared in the orange group spent more time with orange fish versus blue fish, and the fish reared in the blue group spent more time with blue fish versus orange fish. Most of the artificial fish spontaneously developed social preferences for in-group members. **c** To measure the impact of

social experiences in learning social preferences, we reared artificial fish in non-social environments, without other artificial fish. During training, all fish acquired experience with the visual environment, but the fish reared separately did not acquire experience with social partners. Compared to the artificial fish reared in groups, the artificial fish reared separately showed little to no evidence of social grouping in the 2AFC task. The development of social preferences in the artificial fish required experience with social partners, akin to the development of social preferences in newborn fish, who develop preferences for social partners encountered early in life. One-sample $t$-tests were performed to determine statistical significance (*$p < 0.05$, **$p < 0.01$, and ***$p < 0.001$, uncorrected; the exact $p$ values and raw values are provided in the Source Data). Data are presented as mean values. Error bars indicate ±1 SEM.

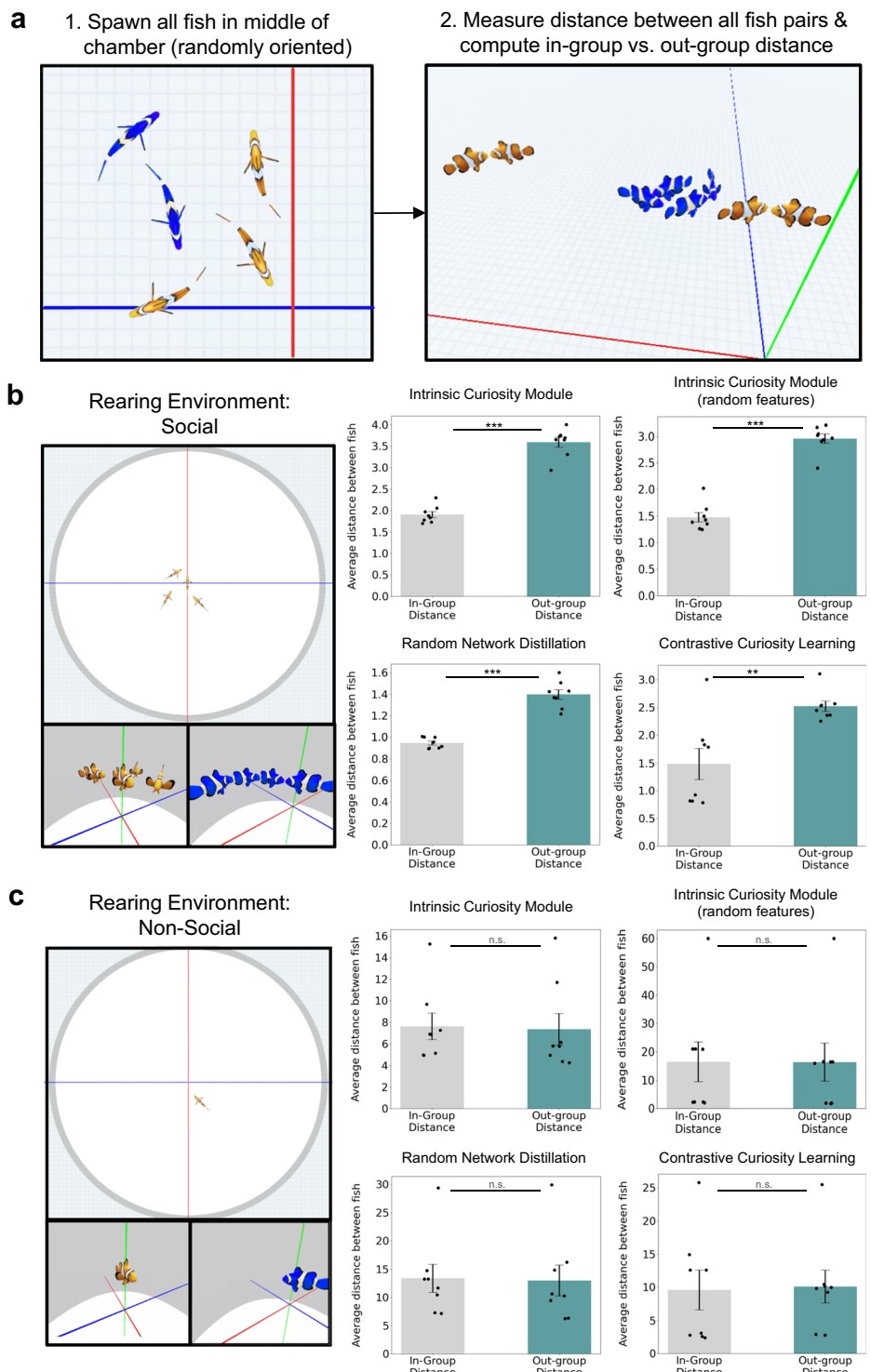

**Fig. 7 | Experiment 3: Self-Segregation Task. a** We also tested the artificial fish ($n = 8$) in a self-segregation task, where we spawned the fish in the middle of the chamber, then measured the distances between all of the fish to measure whether they spontaneously self-segregated into groups. **b** Results on the self-segregation task. The fish reared in the orange group self-segregated with orange fish versus blue fish, and the fish reared in the blue group self-segregated with blue fish versus orange fish. **c** Contrary to the artificial fish reared in groups, the artificial fish reared separately spent similar amounts of time with in-group versus out-group members. Paired samples $t$-tests (two-tailed) were performed to determine statistical significance (*$p < 0.05$, **$p < 0.01$, and ***$p < 0.001$, uncorrected; the exact $p$ values and raw values are provided in the Source Data). Black dots indicate performance of individual artificial fish. Data are presented as mean values. Error bars indicate ±1 SEM.

We then froze the ANNs and tested the social preferences of the artificial fish.

The artificial fish reared separately showed little to no evidence for social preferences in the 2AFC task (Fig. 6c) or self-segregation task (Fig. 7c). The development of social preferences required experience with social partners, akin to the development of social preferences in newborn fish, who develop preferences for social partners seen early in life[50].

## Discussion

What learning algorithms produce social behavior? To address this question, we took a reverse engineering approach. We built ANNs formalizing two core learning principles in psychology and neuroscience (reinforcement learning (RL) and intrinsic motivation), embodied the ANNs in artificial fish, and raised the artificial fish in virtual fish tanks. We then compared the development of social behavior across artificial and biological fish[43,50]. We found that artificial fish spontaneously develop fish-like collective behavior. Like fish, the artificial fish rapidly learned to group from high-dimensional sensory inputs, using intrinsic motivation to guide learning. The artificial fish also learned to turn toward each other early in development, and then increased the intensity of their social interactions over time. This learning trajectory can be captured with simple attraction rules observed in biological fish[43,67].

We also found that artificial fish develop grouping behavior in realistic ocean environments, showing that these embodied models can generalize to naturalistic learning contexts. Finally, we found that artificial fish develop fish-like social preferences. When reared in parallel visual environments as biological fish[50], artificial fish spontaneously develop "us versus them" behavior, preferring in-group over out-group members. These artificial fish self-segregated, spontaneously forming groups with in-group members.

Our study thus reveals a set of core learning algorithms that are sufficient to produce animal-like social behavior in embodied agents (Fig. 8). By revealing learning algorithms that produce animal-like social behavior, our results inform classic questions in biology and psychology. First, *how* do animals learn to group? We show that two generic learning mechanisms, RL and intrinsic motivation, provide a sufficient computational foundation for learning social behavior from raw visual inputs. By optimizing thousands of synaptic weights across thousands of observations to maximize intrinsic information-seeking rewards, artificial agents can rapidly learn to transform high-dimensional sensory inputs into social behavior.

Researchers across psychology[71,72], biology[73], and artificial intelligence[74–76] have long noted that curiosity (and other forms of intrinsic information search) can aid learning. By motivating agents to seek new information and experiences, curiosity provides a generic and universal reward for learning diverse skills. We hypothesize that curiosity produces social behavior because social partners are the least predictable things in a newborn's visual environment. Curiosity-driven learning systems are attracted to unpredictable things and will learn to produce actions that lead to unpredictable outcomes (e.g., actions that keep social agents in view). Curiosity-driven systems should thus develop social behavior when (1) they are raised/trained in environments with social partners (e.g., parents and siblings) and (2) there is a critical period (i.e., cessation of learning), which are widespread in animals, especially during early brain development[77–80]. Under these conditions, we have shown that generic learning systems can rapidly develop core social skills.

Second, these results shed light on *why* animals group. Our developmental simulations show that when embodied agents have RL and intrinsic motivation, the agents will spontaneously develop animal-like social behaviors when raised in environments with social partners. Thus, we provide computationally explicit evidence that collective behavior can develop in the complete absence of hardcoded interaction rules (e.g., cohesion, co-alignment), which are typically used in rule-based models of collective behavior. It is not necessary to hardwire interaction rules into embodied agents to produce collective behavior and social preferences. Rather, generic learning algorithms can drive the development of core social behavior.

The RL and intrinsic motivation algorithms used here were not originally designed to produce social behavior. Nevertheless, when these learning algorithms are embodied and permitted to learn in social environments, social behavior spontaneously develops. Social behavior, with all of its survival benefits, may thus be an emergent property of generic learning algorithms adapting (fitting) to the spatiotemporal statistics of embodied data streams acquired during early postnatal social interactions.

This finding implies that evolution would not have needed to 'discover' innate, domain-specific learning mechanisms to produce social behavior[81,82]. Once evolution discovered generic (i.e., domain-general) learning mechanisms, animals would have rapidly started developing social behavior early in life. We thus speculate that the computational foundations of social behavior are domain-general algorithms (e.g., RL, intrinsic motivation). These generic learning algorithms provide a sufficient computational basis for learning domain-specific social knowledge from experience.

Ultimately, a deep understanding of social behavior requires understanding at multiple levels, ranging from complex, neurally mechanistic models that actually perform the same tasks as animals to simple, low-dimensional models that explain behavior using human-interpretable parameters. To this end, our study links complex neurally mechanistic models (artificial fish) with simple models (attraction rules with a single parameter). This approach illuminates the algorithms driving social behavior, while simultaneously revealing simple attraction rules for understanding common developmental pathways and learning outcomes across biological and artificial systems.

Our study builds on the rich reverse engineering paradigm popularized in systems neuroscience[83–89]. This paradigm aims to discover an engineering-level description of biological intelligence, similar enough to biology to preserve the essential algorithms, but abstract enough to discard details not required for reproducing biological intelligence in artificial systems. A core promise of this paradigm is that it will lead to unified models that explain whole bodies of experimental findings[11]. For example, in the domain of object recognition, image-computable models have been evaluated across a range of images, allowing individual models to be compared across many experimental findings[84]. Engineering-level models are runnable (they perform the same tasks as animals), so these models have the potential to unify large swaths of the scientific literature[83–89].

Our approach has much in common with the reverse engineering approach used to study sensory capacities, including a reliance on precise (high signal-to-noise ratio) data from animals and a shared goal of building neurally mechanistic, image-computable models of biological intelligence. While we did not focus on internal (neural) measurements here, future studies could collect neural measurements and compare developing neural activation patterns across biological and artificial systems.

Our approach also prioritizes different dimensions of the reverse engineering problem. We focus on newborn animals (rather than adult animals) to study the core learning algorithms that drive animal intelligence. We focus on embodied (rather than disembodied) AI models because much of intelligence emerges from an agent's interactions with the world. And we focus on controlled rearing (rather than uncontrolled datasets) to characterize the respective roles of learning machinery and experience on the development of intelligence. Our approach thus extends the call from a large group of scientists arguing for Embodied Turing Tests that involve benchmarking and comparing animals versus machines[90]. We suggest that Newborn Embodied Turing Tests (NETTs)[91,92] will be particularly useful for building unified

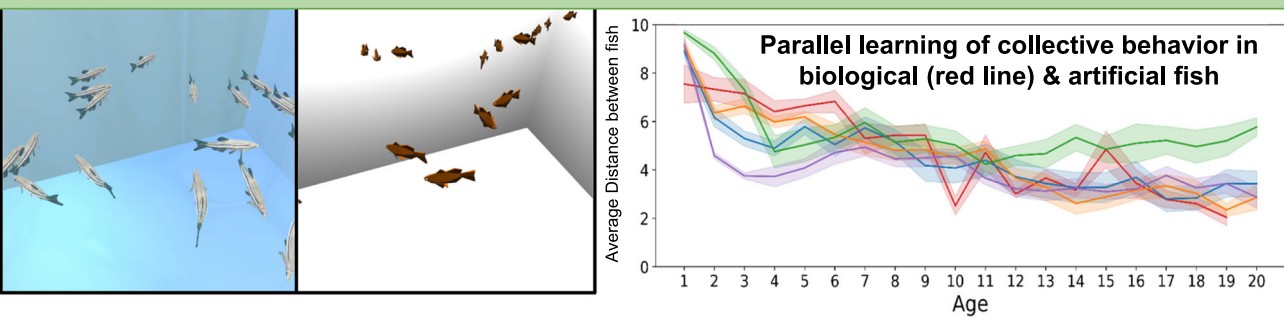

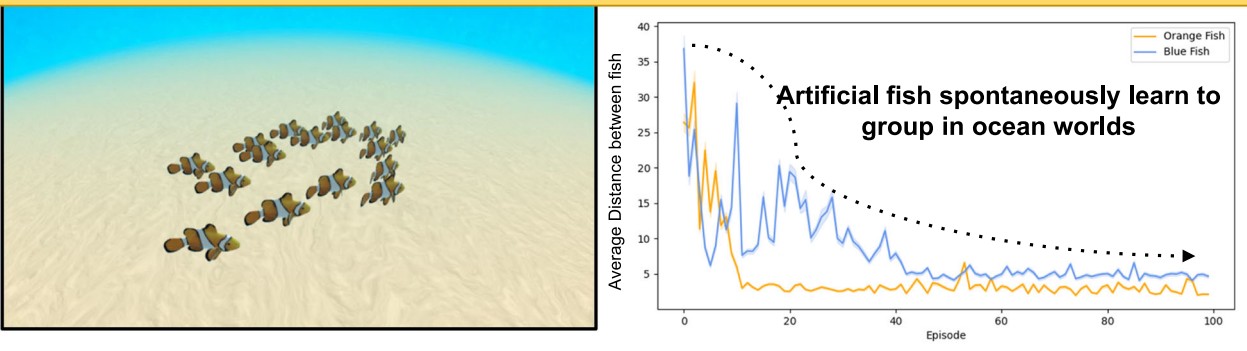

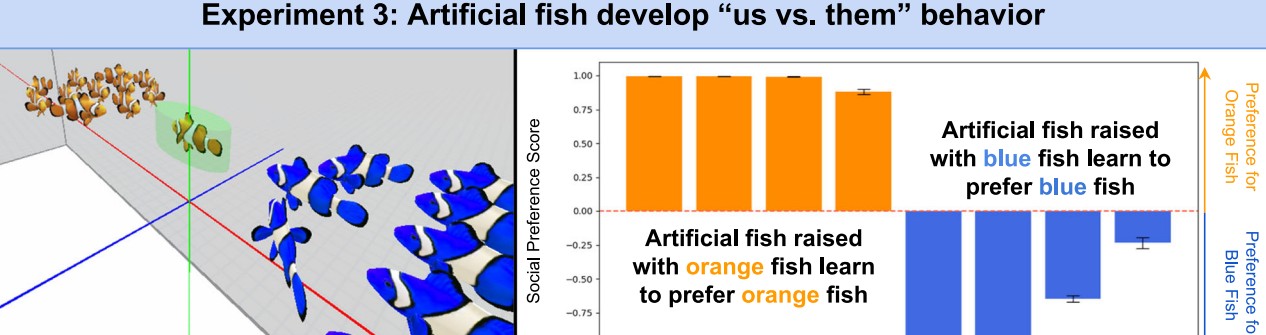

**Fig. 8 | Summary figure.** Across three experiments, we show that biological and artificial fish (embodied deep neural networks) develop the same social behaviors when reared in the same environments. Experiment 1 reports the parallel development of collective behavior in biological and artificial fish; Experiment 2 shows that the artificial fish learn social behavior in naturalistic ocean environments; and Experiment 3 reports the parallel development of social preferences in biological and artificial fish. Our study provides a foundation for reverse engineering the origins and development of social behavior using image-computable models from artificial intelligence.

models of biological intelligence, focused on whether animals and ANNs develop along common pathways and learn similar behaviors over time.

We provide an existence proof that NETTs are tractable and can produce task-performing models of core social behaviors. We emphasize that we did not fit the behavior of the artificial fish to the biological fish in any way. Rather, the artificial fish learned fish-like social behavior when they were reared in similar environments as real fish and had behavioral goals that are ethologically plausible for animals (i.e. curiosity/novelty seeking). Adding biological constraints to the training of ANNs (constraints imposed by brain, body, and environment) led to reasonably accurate models of social behavior. This finding accords with goal-driven modeling studies showing that real-world tasks place strong constraints on the internal parameters of

ANNs[84]. By expanding goal-driven modeling to embodied learning contexts, we can place additional constraints on ANNs and explore the roles of brains, bodies, and environments in the development of intelligent behavior.

Importantly, the body of an animal includes not only its general morphology but also its physiology (e.g., metabolism, thermoregulation, reproductive mechanisms) and sensors (e.g., eyes, ears, nose). If animals have different sensors, for example, then their brains will receive different information. Behavioral differences across animals and ANNs could thus be due to differences in brains, bodies, or environments. All of these factors influence how animals fit to different evolutionary niches.

These results set the stage for many exciting future directions. With the discovery of models that spontaneously develop animal-like social

behavior, we can now search through the model class to find particularly strong models, via a continuous cycle of model creation, model prediction, and model testing against new experimental results (e.g., new controlled-rearing studies). The most promising models can then be investigated in richer detail, leading to greater intuitive understanding of the underlying algorithms. Controlled comparisons with different brains, bodies, and environments could also define the necessary and sufficient conditions for learning animal-like social behavior.

One limitation in building embodied models is the availability of algorithms that can learn with high-dimensional action spaces. To avoid this roadblock, we built artificial fish with low-dimensional action spaces. Future work might close this gap between animals and machines by building more realistic virtual animal bodies, with similar action spaces to animals[93]. This would add additional (and potentially valuable) constraints on ANNs and may help close the behavior gap between animals and machines.

A second limitation is that our artificial fish only had a single objective: to maximize rewards from their intrinsic motivation algorithm. Animals, however, have multiple needs (e.g., hunger, thirst, temperature regulation, curiosity), which likely shape learning and behavior in important ways. This might explain why our artificial fish developed stronger social behavior than biological fish. We suspect that artificial fish would develop weaker (more fish-like) social behavior if they had all of the same needs as biological fish. Future studies could explore this possibility by building artificial fish with fish-like objectives, potentially with control systems that capture the diverse homeostatic needs of biological systems[94].

A third limitation is that although we showed that RL and intrinsic motivation are sufficient to produce social behavior, we have not shown that they are necessary. Other learning algorithms might also produce social behavior in embodied agents. We started with RL and intrinsic motivation because they are deeply rooted in psychological and neuroscience research, but we hope that researchers will explore how other learning algorithms perform in this domain.

In sum, we show that embodied ANNs are viable models of the core learning algorithms that produce social behavior. This digital twin approach lays a foundation for linking pixels-to-actions models in artificial intelligence to the study of social development. Pixels-to-actions models allow researchers to test how brains, bodies, and environments shape the development of social behavior. Researchers can then dissect the best-performing models to understand which components are critical for learning and collect new data to challenge and constrain future generations of social behavior models.

## Methods
### Virtual environments
The virtual environments were created in the Unity Game Engine. Due to its flexibility, Unity is an ideal testbed for many forms of AI simulation. We used a package known as ML-Agents Toolkit[95], which allows researchers to train artificial agents in virtual worlds. We used the following software: Unity ML-Agents version 2.0.1, Python 3.8.10 with PyTorch 1.7.1+cu110, Python mlagents library version 0.26.0, ml-agents-envs version 0.26.0, and ML-agents' Communicator API 1.5.0. The virtual environments mimicked the holding tanks and experimental arenas for raising and testing zebrafish in Hinz & de Polavieja[43] (Experiment 1) and Engeszer et al.[50] (Experiment 3).

In Experiment 1, the virtual rearing tank had white walls, and its floor measured 40 × 40 units. The circular test arena had a radius of 16 units. We trained 20 artificial fish simultaneously in the virtual rearing tank. In Experiment 2, we trained the artificial fish in a realistic underwater seafloor environment with high-resolution sand textures, shadows, drifting ocean particles, and caustic lighting (Fig. 5a). In Experiment 3, we trained the artificial fish in a virtual rearing tank designed to mimic the 100 mL cup described by Engeszner et al.[50].

### Artificial fish
The artificial fish measured 2 units (length) by 0.7 units (height). The artificial fish received visual input through an invisible forward-facing camera attached to its head (64×64 pixel resolution) and a field of view of 160°. Wide fields of view can cause peripheral distortions to cameras, where objects closer to the edges of the visual field appear larger than those at the center. To avoid this, we used Unity's built-in cylindrical (Panini) projection to adjust the agent's camera. This helped mitigate distortions and created a more realistic perspective.

We simulated collisions between the artificial fish using built-in colliders in the Unity game engine. The body of each artificial fish was centered in a capsule collider with a radius of 0.5 units and a length of 2 units, and the position of the artificial fish was defined as the location of the center of its collider. Since two colliders cannot pass through one another, this prevented the artificial fish from moving through one another.

### ANN architecture
We used the same ANN architectures across Experiments 1-3. The policy network contained two convolutional layers with Leaky ReLU activations, followed by two fully connected layers with 128 hidden units and an action output layer. The intrinsic motivation networks had an identical encoder architecture (two convolutional layers with Leaky ReLU activations, followed by two fully connected layers with 128 hidden units).

For the ICM, the inverse dynamics model contained a hidden layer with 256 hidden units and two output heads to predict translation and rotation actions, and the forward dynamics model was a 2-layer MLP with 128 hidden units and 128 output units. The predictor and random network for RND consisted of the convolutional encoder, which produced 128-dimensional representation vectors. The predictor network of CRL consisted of the encoder followed by a projection network, which contained two fully connected layers with 128 hidden units and 128 output units.

### Training the artificial fish
In Experiment 1, the artificial fish could take one full action on every frame of the simulation. One full action was the result of two discrete movement sets: (1) [move fast or move slow] and (2) [rotate left, rotate right, or no rotation]. For example, the fish might decide to [move fast] + [rotate left] on one frame, and then [move fast] + [no rotation] on the next frame. A sharp right turn would then be the result of [move slow] + [rotate right] for several frames. For the artificial fish in Experiments 2-3, one full action was the result of two discrete movement sets: (1) [move forward or stay] and (2) [rotate left, rotate right, or no rotation]. For example, the fish might decide to [move forward] + [rotate left] on one frame, and then [move forward] + [no rotation] on the next frame. A sharp right turn would then be the result of [stay] + [rotate right] for several frames. Each rotation was ~2° along the y axis per step.

At the beginning of each training episode, the positions and orientations of the artificial fish were randomized. The ANN weights in each artificial fish's ANN were randomly initialized at the start of training. We used the Proximal Policy Optimization algorithm (PPO) to update the model weights every five episodes, using a batch size of 500. The reward discount factor was set to 0.99, and the initial learning rate was set to 0.001, with a linear decay schedule. Each episode consisted of 1,000 time steps, and the artificial fish were trained for a total of 2000 episodes (equivalent to 2 million time steps).

In Experiment 2, we trained 16 artificial fish in the virtual ocean world. We trained each fish for one million timesteps. The blue fish and orange fish were trained in separate groups. To reduce the computational load associated with training large numbers of artificial fish simultaneously, we used four brains (ANNs), each of which controlled (and received input from) four fish bodies. This provided the brains

 

with four times as many experiences for learning and allowed us to train larger numbers of fish simultaneously, akin to the large fish shoals found in nature. In Experiment 3, we trained 4 fish in the virtual world modeled after Engeszer et al.[50]. We used the same training approach as in Experiment 1, where each fish was controlled by its own brain (ANN).

Across all experiments, all of the fish in the same group had the same intrinsic motivation algorithm. Each artificial fish started with a different random initialization of connection weights, and each fish's connection weights were shaped by its own particular experiences during training.

### Testing the artificial fish

In Experiment 1, we tested the artificial fish by grouping the 20 trained fish into 10 pairs of fish and then deploying each pair in the test arena. The fish brains (ANNs) were initialized with the saved weights during one of the 20 checkpoints from training and the weights were not updated during the test trials. For each checkpoint, we collected the artificial fish's trajectories for 10,000 time steps in total (2 test trials, each containing 5000 time steps). We then computed the average distance between each pair over the 10,000 time steps and computed the mean across the average distances of the 10 fish pairs.

In Experiment 2, we tested the fish by measuring the average distance between group members across the training episodes. If the fish developed grouping behavior, then the distance between fish should have decreased across the training phase. We observed rapid grouping behavior in both the orange and blue fish (Fig. 5b).

In Experiment 3, we tested the artificial fish using two measures. First, to mimic the testing conditions of the biological fish, we tested the artificial fish using a 2AFC task (Fig. 6a). The chamber contained two shoaling groups ($n = 11$ fish): one group had a familiar pigment (matching the test fish), while the other group had a novel pigment. On each test trial, the test fish was placed in the center of the chamber facing a random direction. On every time step, we recorded the position of the test fish and measured its distance to the center of each shoal. As with biological fish from Engeszer et al.[50], we measured the proportion of time the target fish spent in proximity to the group with the familiar versus novel pigment (measured as the distance to the center of each shoal). Each of the 8 artificial fish were tested separately across 1000 test trials and each test trial contained 3000 actions (steps).

Second, the self-segregation task (Fig. 7a) involved placing all 8 of the trained artificial fish (4 orange fish and 4 blue fish) in a single environment. At the start of each trial, the fish were centered in the environment, oriented randomly, and then allowed to freely move around the environment and interact with other fish. To measure whether the fish self-segregated according to pigment, we measured the Euclidean distance between each fish and every other fish at each time step, then computed the in-group distance (i.e., the average distance to fish of the same color) and the out-group distance (i.e., the average distance to fish of novel color). We tested the fish across 1000 trials, where each trial lasted 3000 time steps (actions).

### Control randomized data

In Experiment 1, following Hinz & de Polavieja[43], we obtained control data for the condition involving pairs of fish by randomizing the data obtained from the experimental procedure. For a test trial of two fish, we paired the position of the first fish at each time step in an episode with the position of the second fish at a randomly selected time step in a different episode (to eliminate correlations). We repeated this process for the second fish, pairing the positions of the second fish with randomized positions of the first fish. We used 20 different random seeds to obtain 20 randomized trials from each original test trial. As a result, for a given checkpoint, we obtained 200 (20 × 10 pairs) randomized trials from the original data.

To compute statistical significance, we followed the approach described by Hinz & de Polavieja[43], where significance was computed as the probability that the control randomized data gave the experimental result. First, we obtained the average distance for each of the 10 fish pairs for each training checkpoint. We then computed the mean across the 10 pairs to obtain $d_{exp}$. Next, we generated 200 randomized trials using the procedure above, and then computed 200 average distance values from the randomized data. We then drew 10 random values from the 200 average distances and computed their mean, d. We repeated this 10,000 times to obtain 10,000 values of d. Finally, we computed the $P$ value as the proportion of these 10,000 values equal to or smaller than the experimental value, $P(d \leq d_{exp})$.

### Measuring turning probabilities

To measure the turning probabilities of the artificial fish in Experiment 1, we generated a large set of test images, each showing a unique spatial configuration of neighbors from the perspective of a focal fish. For each fish configuration (1:0, 2:1, 3:0, etc.), we recorded 600 test images. To create each image, we randomly positioned $N_1$ neighbors on one side and $N_2$ neighbors on the other side of the optical axis of the focal fish's camera. Images were recorded from the camera of a fish whose position and orientation was fixed at the center of the testing arena. Next, we fed the 600 images into the artificial fish's neural network and recorded its output behavior (i.e. actions) in response to each image. Finally, we calculated the frequency of turns towards the sides with $N_1$ versus $N_2$ neighbors to estimate the turning probabilities of the artificial fish.

### Intrinsic motivation algorithms

The intrinsically motivated artificial fish had two learning components: (1) a predictor (reward) network that learned a self-supervised prediction task, and (2) a policy network that learned to convert pixel inputs into action outputs by maximizing the intrinsic reward, which was proportional to the predictor network's error. All of the intrinsic motivation networks produced rewards based on the prediction error of the agent's internal model of the world. Below, we describe the four intrinsic motivation algorithms, each of which employed a different self-supervised task for the predictor network.

**Intrinsic curiosity module (ICM).** ICM uses forward dynamics[56] prediction for its self-supervised task. Instead of making predictions in a high-dimensional sensory space, the model makes predictions in a learned feature space. Specifically, ICM consists of three learning components: a feature encoder $\varphi$, an inverse dynamics model g, and a forward dynamics model f, parametrized by $\theta_\varphi$, $\theta_g$ and $\theta_f$, respectively. Given a transition $(s_t, a_t, s_{t+1})$, where $s_t$ and $s_{t+1}$ are visual observations on consecutive time steps and $a_t$ is an action taken at time t, the visual observations are encoded into representations $x_t = \varphi(s_t; \theta_\varphi)$ and $x_{t+1} = \varphi(s_{t+1}; \theta_\varphi)$ by the feature encoder. The inverse model learns to predict actions $\hat{a}_t = g(x_t, x_{t+1}; \theta_g)$ from the features of two consecutive observations. The forward model learns to predict the features of the observation at time t + 1, $\hat{x}_{t+1} = f(x_t, a_t; \theta_f)$, from the features $x_t$ and action at time t. The prediction error of the forward model is used as the intrinsic reward.

**ICM with random features.** The original formulation of ICM[56] used features learned by an inverse dynamics model for making predictions about the environment's forward dynamics. However, this can result in learning instability since the feature space for prediction continues to change as the inverse dynamics model undergoes training. To fix this issue, Burda et al.[57] eliminated the use of the inverse dynamics model and instead used the feature space of a CNN feature extractor that was randomly initialized and kept fixed thereafter.

**Random network distillation (RND).** RND is a novelty-based approach[58] that uses two neural networks (target network and predictor network). The target network is a neural network that is randomly initialized and then kept fixed, and it defines the self-supervised prediction task. The predictor network is then trained to predict the output of the target network. The prediction error between the output of the target network and the output of the predictor network constitutes the intrinsic reward signal for the policy optimizer. Since the predictor network tends to have lower prediction errors on examples similar to what it has already experienced during training, this intrinsic reward encourages the agents to explore unfamiliar parts of the environment. The target network $f : O \rightarrow \mathbb{R}^k$ transforms an observation x into an embedding, and the predictor network $f' : O \rightarrow \mathbb{R}^k$ is trained to minimize the square error between the prediction and target embeddings $||f'(x; \theta) - f(x)||^2$ with respect to its parameters.

**Contrastive curiosity learning.** We used a self-supervised temporal contrastive[59] learning technique that aims to learn representations of sequential data, such as images. The method involves comparing the representations of different samples in a sequence, where the representations of nearby samples are more similar than those of distant ones. During training, a neural network is trained to minimize a contrastive loss function, which promotes similar representations for nearby (positive) samples and dissimilar representations for distant (negative) samples. Positive pairs are generated by selecting two consecutive observations from the experience buffer, while negative pairs are generated by selecting pairs of observations that are not temporally proximate. The dissimilarity between the representations of two consecutive states is used as the intrinsic reward signal.

### Reporting summary
Further information on research design is available in the Nature Portfolio Reporting Summary linked to this article.

## Data availability
All of the source data used to create the figures in this paper are available as a Source Data file. Source data are provided with this paper.

## Code availability
The data and code used for training and testing artificial fish is available on our GitHub page: https://github.com/buildingamind/McGraw_Lee_Wood_2024.

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

## Acknowledgements
Funded by James McDonnell Foundation Understanding Human Cognition Scholar Award (JNW) and Facebook Artificial Intelligence Award (JNW). We thank Samantha M. W. Wood for comments.

## Author contributions
Conceptualization: J.D.M., D.L., J.N.W., Methodology: J.D.M., D.L., J.N.W., Data Collection: J.D.M., D.L., Data Analysis: J.D.M., D.L., Visualization: J.D.M., D.L., J.N.W., Project administration: J.N.W., Supervision: J.N.W., Writing – original draft: J.N.W., Writing – editing: J.D.M., D.L., J.N.W.

## Competing interests
The authors declare no competing interests.
