## [Peer Review File · Nature Communications]

Parallel development of social behavior in biological and artificial fishREVIEWER COMMENTS

Reviewer #1 (Remarks to the Author):

The authors present a study in which they explore how collective behavior can arise in simulated 'fish' over the course of learning. Their fish are objects in a simple game engine which process visual input and make action selections (moving forward and turning). Multiple fish can be instantiated simultaneously. The objective of learning for each fish is to be maximally curious (i.e., to develop a policy that will choose actions that will result in novel states of the world). Specifically, they employ an 'intrinsic motivation' ANN which generates scalar values that represent how unexpectedly different consecutive states of the world are. These scalars are then used as the rewards needed to use deep RL ANN that yields a behavioral policy. The authors show that over the course of learning, their fish learn to school (i.e., spend more time than expected near other fish in the environment). Overall this is an interesting study that probes the origin of a particular kind of social behavior in agents living in a simulated world with features of real world environments including other agents, sensory inputs, and actions.

Enthusiasm is tempered somewhat by the lack of a clear motivation for the specific setup used which one could argue is perhaps more complex than needed to explore the authors' specific questions or too simple to capture important features of real world behavior.

Additionally, there is an important missing piece (or at least I missed it) motivating the hypothesis that curiosity-driven learning would give rise to schooling. Where does that come from and why is that a reasonable hypothesis? Relatedly, the authors do not explain or discuss why it ultimately works. My guess is that in an environment that is static other than the behavior of other fish, novelty seeking would necessarily result in observing other fish whose movements aren't fully predictable. However, this seems an odd learning objective in naturalistic environments when the most novel things would be, for example, a predator rather than a familiar. I am not terribly familiar with the fish literature, but it would be interesting if there were any studies of schooling with a visual stimulus with unpredictably features projected into the tank. I assume that fish would still school, but I would guess that would not be the case in this model (i.e., they would be attracted to the unpredictably visual stimulus). Despite these reservations, this is an important example of how complex behavior can arise when agents learn to interact with a world.

Specific comments:

1. Why are you using a visual stimulus at all? Given the setup, I would imagine that you'd get the same results if you defined a feature vector as the stimulus (i.e., the fish's current direction, egocentric angles and distances to fish within a viewing window of 160 degrees or whatnot,

distances to walls, etc.). Have you tried this? Is there something specifically different when using a visual input?

2. Relatedly, what are the neural representations that arise in the networks? Perhaps certain importance features are decoded from the visual scene while others aren't. While analysis of ANN activity would be even more appealing in the context of neural data to compare it to, it might still be interesting to characterize the activity to get a handle on the computations that are driving the resultant behavior over learning.

3. You slightly upsell the degree to which you have a fish-like embodied agent, especially in the introduction. Upon first read, I anticipated that you would use a fish body that lives in a world with physics and that moves via an action space somewhat akin to that of a real fish. While you mention that this kind of increased complexity may be interesting for future work in the discussion, you don't motivate the particular level of embodiment that you ultimately use. This is related to the question of whether you need vision. For example, you state that to compare intelligence, you need to raise in silico agents in the same environment as in vivo animals that are doing the same task. One could argue that what you ultimately do is rather far from that aspiration.

4. You say that real fish learn via "self-supervised intrinsic motivation". Is this true?

5. For Fig. 2b, other than for intrinsic curiosity with random features, the results aren't particularly impressive. Should I be surprised that the results in the other figures seem pretty similar across intrinsic motivation algorithms?

6. Fig. 2d for contrastive curiosity looks a little weird with highly non-monotonically changing heatmaps. What gives?

7. Also, Fig. S1 is totally unconvincing and looks nothing like Fig. 2c. Do you have any quantification to back up your claim of similarity? You say "Thus, in some cases, closing the gap between animals and machines may require changing the bodies (rather than brains) of the machines", which I wholeheartedly agree with, but these data don't seem to sell support at all.

8. I'm confused by paragraph that commences with "There was one analysis in which we observed different behavior...". You say "biological fish are more likely to turn toward other fish when there are three fish on one side and zero fish on the other side compared to when there is just one fish on one side and zero fish on the other side". You contrast that with "when there were zero fish on the opposing side, we found that artificial fish were more likely to turn toward three fish than one fish".

Maybe I'm missing something but these sound like the same thing to me. The rest of the paragraph is also rather dense and I failed to parse it. I didn't look up ref 43 however. But frankly, I'd rather not have to.

9. You often talk about needing "intrinsic motivation and deep RL" as the two ingredients here, but it's a bit misleading because the former provides the reward landscape that is needed for the latter. I recommend rephrasing this to say "deep RL with curiosity-derived rewards" or some such. It may be confusing to omit the dependence between the two when you reference them (e.g., in the abstract and introduction).

Small points:

1. The sentence "Psychologists have long recognized that measures of intelligence are task dependent" needs a citation.

2. For Fig. 4a&b: "Blue dots indicate experimental data across successive days/ages." If there is a lawful progression of these dots along the theoretical line as a function of learning, you would be well-served by color-coding the dots by age.

Reviewer #2 (Remarks to the Author):

This paper proposes to extend the "reverse engineering" task-driven AI/biological comparison that has had success in a range of domains into the domain of collective behavior of zebrafish. The authors compare the behaviors of intrinsically motivated reinforcement learners (varying the intrinsic motivation choices) in a virtual environment to zebrafish reared in a controlled environment. The intrinsically motivated RL "artificial fish" learn directly from pixel observations and have no additional reward signal. The authors show a number of behavioral similarities between the biological and artificial fish, namely a similar time evolution of distance between fish as well as similar time evolution of "rules of attraction," which consider the likelihood a fish turns in a given direction as a function of the number of fish on either side of it, as well as other parameters related to such turn probabilities.

This is a really interesting direction, with the potential for broad interest in the domain of computational modeling — to my knowledge, extending this sort of reverse engineering into this sort of social behavioral domain is novel. Previous work has shown that zebrafish exhibit rich

collective behavior with simple regularities; this provides the opportunity to study how such behavior might be emergent in RL agents. The ability to control the rearing of these fish gives modelers the ability to “grow” RL agents in analogously controlled conditions, making this a great starting point for developing reinforcement learning comparisons with biological systems more generally.

The methodology appears sound, with the work supporting the claims of the paper — the authors rightfully frame this as a sort of “existence proof” that intrinsically motivated RL agents given visual experience can exhibit this sort of collective behavior. I do think, though, that the manuscript could be considerably strengthened by giving us a stronger sense of how useful the metrics they put forth are in discerning good models from bad ones. It is not super clear to me how stringent a demand it is on models to exhibit the similarities shown. The primary way I would look to do this would be to include baseline and ablation models which do not exhibit similar qualitative and quantitative similarities. For example, the RL agents could be given simpler objectives (A fixed reward as a function of the number of fish in the field of view? A closeness objective? Intrinsically-motivated agents but with frozen world models?). Clearly some aspects of behavior could not be captured by random policy agents, but it would be informative to see even what happens in that case for some of the statistics computed. The authors do note this limitation in the discussion — that it is not known whether these are necessary learning rules — but I think it would be really helpful to be able to showcase some nontrivial model that does not exhibit these behaviors. As it stands, the only baseline we have to work from is a randomized data baseline, and only for Fig 2. The similarities exhibited in Fig 4 seem particularly striking, but I would feel much more convinced if I was shown that other reasonable dynamical systems/learning algorithms do not exhibit this sort of collective behavior.

My understanding is that the artificial fish, for all experiments presented in the main article, have a single front-facing camera. While they note that this does not produce the same sorts of average neighbor statistics as in the biological fish (Fig 2 c-d), and that a modified artificial fish with adjusted cameras to be more like the eyes of the zebrafish does have more similar average neighbor statistics (Fig S1), they do not proceed to use this modification for the rest of the experiments. Given that these, in my understanding, better match the visual experience of the zebrafish, I found it a confusing choice that they did not include results for the other comparisons with these modified artificial fish. Were the results similar? Or somehow different?

This might not be fairly within scope, as it would require additional biological experiments, but I noticed that the artificial fish had certain “plateauing” behaviors (e.g. Fig 3b, 5b), and I found myself wondering if the biological fish might plateau to similar levels if reared for longer! E.g. Fig 5a shows an impressive exponential curve, which has some similarities to artificial fish (albeit on different timescales depending on the intrinsic motivation), but some artificial fish are basically at ceiling $p_s \sim 1$, whereas others are not. What happens in the biological case?

Minor: I might be mis-parsing this, but when describing the results of Fig S2 in the main text (end of page 9), the authors say that the biological fish are “more likely” to turn in the 3:0 case vs the 1:0 case, whereas I think what they mean to say (at least, that’s my understanding of Fig S2) is that the biological fish are equally likely to turn in these cases.

Best,

Nick Haber

Reviewer #3 (Remarks to the Author):

Review of NCOMMS-23-22656

I applaud the authors on an excellent, highly compelling submission that is beautifully motivated, empirically sound, and thoughtfully executed throughout the vast majority of the work. I believe that with a few key methodological clarifications, and some reconsideration and reworking of the current results figures, that this piece merits publication in Nature Communications.

Below are my major comments:

The most significant modifications necessary to the current manuscript are modifications that clarify how the current patterns of results are meaningful products of the curiosity-driven, stimulus-computing, embodied-agent learning that the authors are most directly championing, and not simply the trivial outputs of what are otherwise simple reinforcement-learning, ideal-observer models that the authors might consider ‘rule-based’ models of collective behavior.

The first of these clarifications is to the kinds of visual signal that the artificial animal can access directly from its sensory camera. In short, what kind of learning is really happening here and how can the authors provide evidence of this learning? Can the camera be reduced to a motion detector that somewhat automatically learns to parse the visual array into ‘fish-not-fish’? Part and parcel of embodied visual learning in real-world agents are visual systems that by necessity must learn a set of invariances and selectivities that preserve the ability to recognize ecologically important stimuli under various kinds of external conditions, while also differentiating one ecologically relevant stimulus (e.g. food) from another (e.g. a conspecific). As far as I was able to understand from the

manuscript – in other words, unless I missed something major – there’s only one kind of stimulus in the artificial environment – that is, other fish. I realize this environment is meant to serve as a direct artificial instantiation of the environment used in the target behavioral experiment, but its simplicity makes me think that any visual learning that takes place (at least for the artificial agents) is trivial. Some sort of demonstration the authors could provide, showing perhaps that the visual system of the agents is capable of learning not just the presence or absence of other fish in the visual array, but the orientation of those fish, would go a long way in ensuring the visual learning component of this setup is not otherwise trivial. (In other words, why tout an artificial visual system if it effectively reduces to a motion detector)?

The second clarification is for the reinforcement learning signal. Again, if my understanding is correct, the ONLY signal that is being optimized for the artificial fish is the curiosity-style meta-learning loss. This seems ecologically unrealistic, since biological fish will be motivated not only by their meta-learning, but by the search for food and mates. Are we then to believe that the collective behavior we observe in fish is orthogonal to these additional reward signals? The authors do make mention of this in their discussion on limitations, but this is an ambiguity should in my opinion be addressed far earlier in the setup for the experiment. Again, the reason for this is because an argument of sufficiency alone is not exactly what I would consider solid ground for the claims the authors make about the advantage of their curiosity-driven, embodied-learning agents, since rule-based approaches have for some time given us equally sufficient learning rules that explain collective behavior equally well. This is not at all to say that I myself am not compelled by the argument of sufficiency, but if this alternative approach (which is often much more complicated than the rule-based approach) is to be taken seriously, it should likely go above and beyond in explaining certain phenomena (or providing more ecologically granular predictions) than the rule-based approach.

To summarize my first set of concerns, then, my sense is that that neither the visual learning nor the reinforcement learning in this setup is challenging enough to empirically tout its advantage over a rule-based approach. The visual learning may effectively just be motion detection, and the reinforcement-learning requires nowhere near the multivariate complexity that ethologically realistic reinforcement-learning inherently entails. The argument the authors are making, then, reduces to an argument from first principles: Their approach is better because it uses raw sensory inputs and embodiment. I WHOLEHEARTEDLY agree, but for this to be more than a philosophy paper, the empirical results should make it clear why this kind of learning does not ultimately reduce to the very same rule-learning the authors seem keen on challenging. I welcome any and all additional data or supplementary arguments the authors can muster to change my mind here. (I again also note that this comment is not meant to negate what I perceive to be the deep value of this paper; only to make it much stronger as a counterpoint to what already exists).

Now, for my second major concern: Figures 2-3-4-5. It is exceedingly difficult to grok the main gist of the results from these figures while constantly moving my eyes back and forth from the biological

animal reference to the artificial animal comparisons. Why not reprint the biological animal results on each of the subplots? It appears both the reference and the targets use the same scale in almost all comparisons (except in figure 5, but why not there as well?), so it seems pretty straightforward to combine them. For what it's worth, I think each result could actually be one single plot: with colors for the reference and targets, and linetypes (solid – dashed) for the actual data versus randomized control). It would also help to include a summary figure that uses some sort of fit statistic or distance metric to illustrate how far each model is from the reference. A nitpicky point, but many of the figures in their current form are also low-resolution, making it difficult to read axes and key bits of annotating text. These figures must in my opinion be improved prior to the manuscript being accepted.

In terms of minor comments:

The introductory literature review on the current state of DNN modeling for brains and behavior alike is somewhat oversimplistic and outdated. It is no longer the case, for example, that object recognition models are the dominant force in this kind of modeling – and in fact, most recent work in these domains almost exclusively leverages self-supervised (albeit disembodied) models. In terms of self-supervised embodied models in mice and fish, the works of Josh Merel and Bence Olvecky seem highly relevant here, but are not otherwise cited.

Figure 5: The curve for the biological fish is the only truly exponential curve in this comparison; the rest (for the artificial fish) seem pretty sigmoidal. It does not appear to be the case that the authors meaningfully address this, but they probably should.

Response to Reviewers

Revised manuscript for *Nature Communications*
Parallel development of collective behavior and social preferences in fish and machines
[NCOMMS-23-22656A]

Response to Reviewer #1

The authors present a study in which they explore how collective behavior can arise in simulated 'fish' over the course of learning. Their fish are objects in a simple game engine which process visual input and make action selections (moving forward and turning). Multiple fish can be instantiated simultaneously. The objective of learning for each fish is to be maximally curious (i.e., to develop a policy that will choose actions that will result in novel states of the world). Specifically, they employ an 'intrinsic motivation' ANN which generates scalar values that represent how unexpectedly different consecutive states of the world are. These scalars are then used as the rewards needed to use deep RL ANN that yields a behavioral policy. The authors show that over the course of learning, their fish learn to school (i.e., spend more time than expected near other fish in the environment). Overall this is an interesting study that probes the origin of a particular kind of social behavior in agents living in a simulated world with features of real world environments including other agents, sensory inputs, and actions.

Enthusiasm is tempered somewhat by the lack of a clear motivation for the specific setup used which one could argue is perhaps more complex than needed to explore the authors' specific questions or too simple to capture important features of real world behavior.

The Reviewer raises an important point: in the original submission, we did not clearly explain why we chose to model collective behavior using pixels-to-actions models. Our revision addresses this issue in two ways. First, our revision now explains why we chose pixels-to-actions models, including the following text (pg. 5):

Unified scientific models should explain findings across many studies. This is one reason why pixels-to-actions models are valuable from a scientific perspective. Pixels-to-actions models can be reared and tested in the same environments as animals, so they can be directly compared to animals across a wide range of studies. We can select other studies from the literature, run digital twin experiments, then test whether the same model learns the same behaviors as animals across studies. This idea embraces the integrative benchmarking approach used to reverse engineer the ventral visual stream¹¹, but expands the approach to embodied learning contexts.

Second, to show how pixels-to-actions models can unify diverse findings, we added two new experiments to the paper:

Experiment 2 (pg. 5) shows that the artificial fish from Experiment 1 can develop collective behavior in naturalistic ocean worlds, showing that these embodied models can generalize to real-world learning contexts. **Experiment 3** (pg. 5-6) shows that when these artificial fish are reared in parallel visual environments as biological fish, the artificial fish also develop “us vs. them” behavior, akin to biological fish. These experiments show that single models can account for the development of multiple social skills, including collective behavior (Experiment 1), social learning in complex visual environments (Experiment 2), and social preferences (Experiment 3). We thus argue that pixels-to-actions models can serve as unified models of the core learning algorithms that drive social

behavior in newborn animals: any learning situation given to newborn animals can also be given to artificial agents. This forms a closed-loop system between the study of biological and artificial intelligence (illustrated in Fig. 1). With a closed-loop system, we can test the sufficiency of brains, bodies, and environments for producing social behavior, dissect the best-performing models to understand which components are critical for learning, and collect new data to challenge and constrain future generations of social behavior models (pg. 8).

Additionally, there is an important missing piece (or at least I missed it) motivating the hypothesis that curiosity-driven learning would give rise to schooling. Where does that come from and why is that a reasonable hypothesis?

As suggested, we added more details motivating the hypothesis that curiosity-driven learning gives rise to schooling and social behavior. First, we explain our hypothesis in more detail (pg. 3):

Intrinsic motivation drives learning in humans and animals⁵⁸, including fish⁵⁹, so we hypothesized that intrinsic motivation and RL would drive the development of social behavior in machines. Social partners are typically the least predictable things in a newborn's visual environment, so embodied agents equipped solely with intrinsic motivation and generic learning algorithms (e.g., deep RL) should learn to track and follow social partners. If our hypothesis is accurate, then artificial fish should develop common social behaviors as biological fish when artificial fish are equipped with ANNs that learn solely through deep RL and intrinsic motivation.

Second, in the Discussion, we provide more context into why social grouping might be expected to emerge from deep RL with curiosity-derived rewards (pg. 6):

Researchers across psychology^{70,71}, biology⁷², and artificial intelligence⁷³⁻⁷⁵ have long noted that curiosity (and other forms of intrinsic information search) can aid learning. By motivating agents to seek new information and experiences, curiosity provides a generic and universal reward for learning diverse skills. We hypothesize that curiosity produces social behavior because social partners are the least predictable things in a newborn's visual environment. Curiosity-driven learning systems are attracted to unpredictable things and will learn to produce actions that lead to unpredictable outcomes (e.g., actions that keep social agents in view). Curiosity-driven systems should thus develop social behavior when (1) they are raised/trained in environments with social partners (e.g., parents and siblings) and (2) there is a critical period (i.e., cessation of learning), which are widespread in animals, especially during early brain development⁷⁶⁻⁷⁹. Under these conditions, we have shown that generic learning systems can rapidly develop core social skills.

Relatedly, the authors do not explain or discuss why it ultimately works. My guess is that in an environment that is static other than the behavior of other fish, novelty seeking would necessarily result in observing other fish whose movements aren't fully predictable. However, this seems an odd learning objective in naturalistic environments when the most novel things would be, for example, a predator rather than a familiar. I am not terribly familiar with the fish literature, but it would be interesting if there were any studies of schooling with a visual stimulus with unpredictably features projected into the tank. I assume that fish would still school, but I would guess that would not be the case in this model (i.e., they would be attracted to the unpredictably visual stimulus). Despite these reservations, this is an important example of how complex behavior can arise when agents learn to interact with a world.

The Reviewer raises excellent points and wonders whether this kind of model could actually learn in more naturalistic environments. Our two new experiments address these concerns. First, in

Experiment 2, we explored whether the artificial fish from Experiment 1 would still show collective behavior when trained in more realistic ocean environments, akin to those faced by fish in nature. We first created a realistic virtual underwater seafloor world with high-resolution sand textures, shadows, drifting ocean particles, and caustic lighting. During training, we then measured the average pairwise distance across all of the fish in the group. If the artificial fish can develop collective behavior in this naturalistic world, then the average distance between fish should decline across the training period. All four intrinsic motivation algorithms rapidly developed collective behavior (Fig. 3B). This experiment shows that these artificial fish can develop collective behavior in naturalistic visual worlds, confirming that these embodied models generalize to real-world learning contexts.

Second, in **Experiment 3**, we show that the artificial fish used in Experiment 1 also develop fish-like social preference. Thus, across two controlled environments, we show parallel learning patterns across biological and artificial fish.

Third, we emphasize that pixels-to-actions models can generate predictions for new fish experiments (i.e., by running the artificial fish in digital replicas of the environments planned for fish experiments). If a model is accurate, then it should develop common social behaviors as real fish when reared in the same environments. Pixels-to-actions models can thus be falsified in a straightforward manner.

Finally, returning to the Reviewer's main point, we predict that newborn fish *would* imprint on unpredictable things, regardless of whether those things are caregivers versus predators. In the real world, the early visual experience of most animals consists mostly of caregivers (not predators). We speculate that this early visual experience with caregivers and groupmates naturally gives rise to social behavior. The digital twin approach we introduce in our paper provides an ideal testing ground for testing this (and other) predictions in future studies.

Specific comments:

1. Why are you using a visual stimulus at all? Given the setup, I would imagine that you'd get the same results if you defined a feature vector as the stimulus (i.e., the fish's current direction, egocentric angles and distances to fish within a viewing window of 160 degrees or whatnot, distances to walls, etc.). Have you tried this? Is there something specifically different when using a visual input?

It was necessary to use visual input to build pixels-to-actions models of social behavior. Since pixels-to-actions models can be raised and tested in the same environments as animals, they can serve as unified models of animal intelligence. The revised manuscript clarifies this important point.

2. Relatedly, what are the neural representations that arise in the networks? Perhaps certain importance features are decoded from the visual scene while others aren't. While analysis of ANN activity would be even more appealing in the context of neural data to compare it to, it might still be interesting to characterize the activity to get a handle on the computations that are driving the resultant behavior over learning.

We agree that this would be an exciting future direction. While our current setup does not allow for neural recordings, we hope other researchers will build on our approach to incorporate neural recordings. To allow researchers to easily extend our existing work, we will make all of our models and virtual environments publicly available.

3. You slightly upsell the degree to which you have a fish-like embodied agent, especially in the introduction. Upon first read, I anticipated that you would use a fish body that lives in a world with physics and that moves via an action space somewhat akin to that of a real fish. While you mention that this kind of increased complexity may be interesting for future work in the discussion, you don't motivate

the particular level of embodiment that you ultimately use. This is related to the question of whether you need vision. For example, you state that to compare intelligence, you need to raise in silico agents in the same environment as in vivo animals that are doing the same task. One could argue that what you ultimately do is rather far from that aspiration.

The Reviewer raises a great point: What level of embodiment and task is sufficient to reproduce biological intelligence? One possibility is that simple action spaces will suffice, and that what really matters is allowing neural networks to choose their own inputs during learning (regardless of whether the agents have a simple versus complex action space). Alternatively, motor complexity might be essential for robust animal-like intelligence. As we note in the Discussion, we see this as an important question for future research.

Ultimately, we chose a more simple action space because (1) it was computationally tractable given existing machine learning models and (2) the reverse engineering paradigm often progresses by starting with the most simple models, pushing them as far as they can go, and then adding in additional complexity only when it's necessary to account for a phenomena. This allows researchers to discover which details matter, and which do not, for reproducing biological behaviors.

Finally, inspired by the Reviewer's critique, our lab has started exploring whether these findings generalize to more complex artificial animal bodies. We now have preliminary evidence that artificial fish with more complex bodies (e.g., "rag doll fish" with larger action spaces) can successfully learn to swim and produce collective behavior. This new work is not mature enough to include in the present paper, but we wanted to let the Reviewer know that we agree it is an important direction.

4. You say that real fish learn via "self-supervised intrinsic motivation". Is this true?

Fish do show evidence for intrinsic motivation, which does not come from an external source. We now cite a study providing evidence for intrinsic motivation in fish (pg. 3).

5. For Fig. 2b, other than for intrinsic curiosity with random features, the results aren't particularly impressive. Should I be surprised that the results in the other figures seem pretty similar across intrinsic motivation algorithms?

Thank you for pointing out the lack of clarity in this figure. The key result is that for both biological and artificial fish, the mean distance between fish decreases across the training phase. In our original submission, we presented the fish data side-by-side with randomized control data. We now realize that this was a confusing way to present the data, since it encourages readers to focus on the difference between the fish data and the randomized control data, rather than the key result (the distance between fish decreased across the training phase: the signature of collective behavior). To make this figure more clear, we revised the figure so that the biological and artificial fish data are shown on the same graph and the randomized control data are removed. In Fig. 2a, readers can now clearly see that for both biological and artificial fish, the distance between fish decreased across the training phase. For completeness, we also show the original randomized control data in Fig. S1. We also added a video (SI Video 1) to show the grouping behavior of the fish across the training phase.

6. Fig. 2d for contrastive curiosity looks a little weird with highly non-monotonically changing heatmaps. What gives?

The Reviewer correctly points out that the artificial fish endowed with contrastive curiosity show a non-monotonically changing pattern of collective behavior. This pattern can also be seen in the line graph shown in Fig. 2a, where the mean distance between artificial fish decreases, but then increases,

across the training phase. One of our goals in including different intrinsic motivations in our experiments is to demonstrate that different algorithms can produce different patterns of development. Since fish do not (to our knowledge) show a non-monotonically changing pattern of collective behavior, we speculate that contrastive curiosity is not as accurate of a model of fish social behavior as the other three models.

7. Also, Fig. S1 is totally unconvincing and looks nothing like Fig. 2c. Do you have any quantification to back up your claim of similarity? You say "Thus, in some cases, closing the gap between animals and machines may require changing the bodies (rather than brains) of the machines", which I wholeheartedly agree with, but these data don't seem to sell support at all.

We agree and have removed this argument from the paper.

8. I'm confused by the paragraph that commences with "There was one analysis in which we observed different behavior...". You say "biological fish are more likely to turn toward other fish when there are three fish on one side and zero fish on the other side compared to when there is just one fish on one side and zero fish on the other side". You contrast that with "when there were zero fish on the opposing side, we found that artificial fish were more likely to turn toward three fish than one fish". Maybe I'm missing something but these sound like the same thing to me. The rest of the paragraph is also rather dense and I failed to parse it. I didn't look up ref 43 however. But frankly, I'd rather not have to.

Thank you for catching this mistake. This error has been corrected (pg. 4).

9. You often talk about needing "intrinsic motivation and deep RL" as the two ingredients here, but it's a bit misleading because the former provides the reward landscape that is needed for the latter. I recommend rephrasing this to say "deep RL with curiosity-derived rewards" or some such. It may be confusing to omit the dependence between the two when you reference them (e.g., in the abstract and introduction).

We agree and now refer to the algorithm as "deep RL with curiosity-derived rewards."

Small points:

1. The sentence "Psychologists have long recognized that measures of intelligence are task dependent" needs a citation.

As suggested, we now cite three classic papers supporting this claim (pg. 2).

2. For Fig. 4a&b: "Blue dots indicate experimental data across successive days/ages." If there is a lawful progression of these dots along the theoretical line as a function of learning, you would be well-served by color-coding the dots by age.

As suggested, we now color code the dots in this graph by age (Fig. 2f and Fig. S3).

Response to Reviewer #2

This paper proposes to extend the "reverse engineering" task-driven AI/biological comparison that has had success in a range of domains into the domain of collective behavior of zebrafish. The authors compare the behaviors of intrinsically motivated reinforcement learners (varying the intrinsic motivation choices) in a virtual environment to zebrafish reared in a controlled environment. The intrinsically motivated RL "artificial fish" learn directly from pixel observations and have no additional reward signal. The authors show a number of behavioral similarities between the biological and artificial fish, namely a similar time evolution of distance between fish as well as similar time evolution of "rules of attraction,"

which consider the likelihood a fish turns in a given direction as a function of the number of fish on either side of it, as well as other parameters related to such turn probabilities.

This is a really interesting direction, with the potential for broad interest in the domain of computational modeling — to my knowledge, extending this sort of reverse engineering into this sort of social behavioral domain is novel. Previous work has shown that zebrafish exhibit rich collective behavior with simple regularities; this provides the opportunity to study how such behavior might be emergent in RL agents. The ability to control the rearing of these fish gives modelers the ability to “grow” RL agents in analogously controlled conditions, making this a great starting point for developing reinforcement learning comparisons with biological systems more generally.

The methodology appears sound, with the work supporting the claims of the paper — the authors rightfully frame this as a sort of “existence proof” that intrinsically motivated RL agents given visual experience can exhibit this sort of collective behavior. I do think, though, that the manuscript could be considerably strengthened by giving us a stronger sense of how useful the metrics they put forth are in discerning good models from bad ones. It is not super clear to me how stringent a demand it is on models to exhibit the similarities shown. The primary way I would look to do this would be to include baseline and ablation models which do not exhibit similar qualitative and quantitative similarities. For example, the RL agents could be given simpler objectives (A fixed reward as a function of the number of fish in the field of view? A closeness objective? Intrinsically-motivated agents but with frozen world models?). Clearly some aspects of behavior could not be captured by random policy agents, but it would be informative to see even what happens in that case for some of the statistics computed. The authors do note this limitation in the discussion — that it is not known whether these are necessary learning rules — but I think it would be really helpful to be able to showcase some nontrivial model that does not exhibit these behaviors. As it stands, the only baseline we have to work from is a randomized data baseline, and only for Fig 2. The similarities exhibited in Fig 4 seem particularly striking, but I would feel much more convinced if I was shown that other reasonable dynamical systems/learning algorithms do not exhibit this sort of collective behavior.

The Reviewer makes a great point. Ideally, these experiments could reveal which model features matter—and which do not—for reproducing animal-like social grouping. To address this critique, we made two changes in the paper.

First, we added a new experiment (Experiment 3) focused on the development of social preferences. Specifically, we tested whether the core learning algorithms used in Experiment 1 are sufficient to develop social preferences seen in biological fish. If so, then the artificial fish should spontaneously develop “us vs. them” behavior, learning to prefer members of their group over members of other groups. We found that when artificial fish were reared in parallel visual environments as biological fish, the artificial fish spontaneously developed social preferences, preferring in-group over out-group members. We thus argue that pixels-to-actions models can serve as unified models of the learning algorithms that drive core social behaviors in newborn animals: whatever learning situation researchers give newborn animals can also be given to artificial agents. As illustrated in Fig. 1, this forms a closed-loop system between the study of biological and artificial intelligence, which is often regarded as a gold standard in science and engineering for distinguishing between candidate models.

We also performed ablation experiments to test which components of the model and environment matter for producing social behavior. To test whether intrinsic motivation was important for the development of social preferences in artificial fish, we created a new batch of artificial fish and reduced the strength of the curiosity reward from 1.0 to .001. We made no other changes to the experiment. Contrary to the curiosity-driven artificial fish, we found little to no evidence of social preferences in the curiosity-ablated fish in the 2AFC task (pg. 6, Fig. S4a). We also found that the curiosity-ablated fish failed to self-segregate, spending equal amounts of time with in-group vs. out-group fish (Fig. S4b). In the absence of a robust curiosity-derived reward landscape to drive learning, the artificial fish did not develop social preferences.

Next, we examined the necessity of social experiences on the development of grouping preferences in artificial fish (pg. 6, Fig. 5). In the experiment described above, the artificial fish were reared in groups with other fish (akin to animals, who are reared in groups with siblings). To explore whether this early social experience is necessary to develop social grouping, we reared a second group of artificial fish in the same environment, but without social partners (other artificial fish). During training, these agents acquired experience with the visual environment, but did not acquire experience with social partners (Fig. 5a). After training, we then froze learning in the artificial fish and tested their imprinting behavior in a group (like the fish who were reared together).

The artificial fish reared separately showed little to no evidence for social preferences in the 2AFC task (Fig. 5b). The artificial fish reared separately also showed no evidence of self-segregation behavior, spending equal amounts of time with in-group vs. out-group members (Fig. 5c). This result suggests that the development of social grouping in artificial fish requires visual experience with social partners, akin to the development of social preferences in newborn animals, who develop preferences for social partners encountered early in life. Thus, we reveal two ways in which these models fail to show grouping behavior (ablating curiosity and removing social partners from the visual diet).

My understanding is that the artificial fish, for all experiments presented in the main article, have a single front-facing camera. While they note that this does not produce the same sorts of average neighbor statistics as in the biological fish (Fig 2 c-d), and that a modified artificial fish with adjusted cameras to be more like the eyes of the zebrafish does have more similar average neighbor statistics (Fig S1), they do not proceed to use this modification for the rest of the experiments. Given that these, in my understanding, better match the visual experience of the zebrafish, I found it a confusing choice that they did not include results for the other comparisons with these modified artificial fish. Were the results similar? Or somehow different?

Reviewer #1 had a similar concern about the two-eyed fish data that we reported in the paper, so we removed this argument. We focused on one-eyed fish in this paper for simplicity (i.e., two-eyed visual processing requires more complex visual architectures than one-eyed visual processing). That said, closing the behavior gap between animals and machines might require changing both the brains and bodies of the machines. Thus, to allow researchers to test whether two-eyed fish show even more fish-like social behavior than one-eyed fish, we modified our benchmark so researchers can select either a “one-eyed” or “two-eyed” fish. For the “two-eyed” fish, the benchmark will output two images on each time step, one from each of the two cameras, which are positioned on each side of the fish’s head (akin to the placement of eyes on fish). Researchers can then build and test their own binocular artificial fish, allowing for systematic exploration of the computational role of one vs. two eyes on the development of social behavior.

This might not be fairly within scope, as it would require additional biological experiments, but I noticed that the artificial fish had certain “plateauing” behaviors (e.g. Fig 3b, 5b), and I found myself wondering if the biological fish might plateau to similar levels if reared for longer! E.g. Fig 5a shows an impressive exponential curve, which has some similarities to artificial fish (albeit on different timescales depending on the intrinsic motivation), but some artificial fish are basically at ceiling $p_s \sim 1$, whereas others are not. What happens in the biological case?

This is a good suggestion. While the biological fish were not reared for longer than 24 days, the authors did test adult fish, which had P_s values (attraction rates) of 0.54. When this adult data are added to the graph of the biological fish, the pattern of development looks similar to the artificial fish (i.e., we see the “plateauing” behavior that the Reviewer predicts). To make this developmental commonality more clear, we revised the graph showing the biological and artificial fish (Fig. 2e).

Minor: I might be mis-parsing this, but when describing the results of Fig S2 in the main text (end of page 9), the authors say that the biological fish are “more likely” to turn in the 3:0 case vs the 1:0 case, whereas I think what they mean to say (at least, that’s my understanding of Fig S2) is that the biological fish are equally likely to turn in these cases.

Thank you for catching this mistake. This error has been corrected (pg. 4).

Response to Reviewer #3

I applaud the authors on an excellent, highly compelling submission that is beautifully motivated, empirically sound, and thoughtfully executed throughout the vast majority of the work. I believe that with a few key methodological clarifications, and some reconsideration and reworking of the current results figures, that this piece merits publication in Nature Communications.

Below are my major comments:

The most significant modifications necessary to the current manuscript are modifications that clarify how the current patterns of results are meaningful products of the curiosity-driven, stimulus-computing, embodied-agentic learning that the authors are most directly championing, and not simply the trivial outputs of what are otherwise simple reinforcement-learning, ideal-observer models that the authors might consider ‘rule-based’ models of collective behavior.

The first of these clarifications is to the kinds of visual signals that the artificial animal can access directly from its sensory camera. In short, what kind of learning is really happening here and how can the authors provide evidence of this learning? Can the camera be reduced to a motion detector that somewhat automatically learns to parse the visual array into ‘fish-not-fish’? Part and parcel of embodied visual learning in real-world agents are visual systems that by necessity must learn a set of invariances and selectivities that preserve the ability to recognize ecologically important stimuli under various kinds of external conditions, while also differentiating one ecologically relevant stimulus (e.g. food) from another (e.g. a conspecific). As far as I was able to understand from the manuscript – in other words, unless I missed something major – there’s only one kind of stimulus in the artificial environment – that is, other fish. I realize this environment is meant to serve as a direct artificial instantiation of the environment used in the target behavioral experiment, but its simplicity makes me think that any visual learning that takes place (at least for the artificial agents) is trivial. Some sort of demonstration the authors could provide, showing perhaps that the visual system of the agents is capable of learning not just the presence or absence of other fish in the visual array, but the orientation of those fish, would go a long way in ensuring the visual learning component of this setup is not otherwise trivial. (In other words, why tout an artificial visual system if it effectively reduces to a motion detector)?

The Reviewer raises an excellent point. As discussed above, we used a pixels-to-actions model because this level of model allows us to raise and test newborn animals and machines in the same learning settings. Thus, we can run new experiments to test whether, for example, the artificial visual systems effectively reduce to motion detectors. To this end, the revised paper contains a new experiment (Experiment 3) focused on the development of social preferences. We focused on social preferences for two reasons. First, collective behavior and social preferences are typically studied separately. For instance, the rule-based models used to study collective behavior are not commonly used to study social preferences, since rule-based models have hardcoded interaction rules, whereas social preferences are widely thought to be learned. Studying collective behavior and social preferences with the same pixels-to-actions model provides an opportunity to build bridges between these fields at an engineering-level of abstraction (the level at which we build models that learn like animals). Second, social preferences are widespread across the animal kingdom. Many animals, including humans, develop social preferences early in life, rapidly learning to favor “us” over “them” during social interactions. However, despite significant interest in social preferences across psychology and neuroscience, we know little about the core learning algorithms that drive social

preferences in humans and animals. What learning algorithms—present in newborn animals—cause social preferences to develop so rapidly and flexibility early in life?

To tackle this question, we tested whether the core learning algorithms used in Experiment 1 are sufficient to develop social preferences seen in biological fish. If so, then the artificial fish should spontaneously develop “us vs. them” behavior, learning to prefer members of their group over members of other groups. We found that when artificial fish were reared in parallel visual environments as biological fish, the artificial fish spontaneously developed social preferences, preferring in-group over out-group members. Like real fish, artificial fish learn color-based social preferences (i.e., “us” vs. “them”) based on early visual experiences. These color-based visual preferences show that their visual systems go beyond mere motion detectors.

The Reviewer also discusses embodied agents needing to learn a set of invariances and selectivities that preserve the ability to recognize ecologically important stimuli under various kinds of external conditions. To show that our artificial fish learned some invariances, we added a new experiment (Experiment 2) in which we explored whether the artificial fish from Experiment 1 would still show collective behavior when trained in more realistic ocean environments, akin to those faced by fish in nature. We first created a realistic virtual underwater seafloor world with high-resolution sand textures, shadows, drifting ocean particles, and caustic lighting. During training, we then measured the average pairwise distance across all of the fish in the group. If the artificial fish can develop collective behavior in this naturalistic world, then the average distance between fish should decline across the training period. All four intrinsic motivation algorithms rapidly developed collective behavior (Fig. 3b). This experiment shows that these artificial fish can develop collective behavior in naturalistic visual worlds, confirming that these embodied models can generalize to real-world learning contexts.

Finally, we emphasize in the revision that our experiments will be released as developmental benchmarks for other researchers. Researchers will be able to plug any ML algorithm into the artificial fish and train those fish in the same visual environments as biological fish. We hope that these benchmarks will help drive a community-wide effort to reverse engineer the learning algorithms driving social behavior.

The second clarification is for the reinforcement learning signal. Again, if my understanding is correct, the ONLY signal that is being optimized for the artificial fish is the curiosity-style meta-learning loss. This seems ecologically unrealistic, since biological fish will be motivated not only by their meta-learning, but by the search for food and mates. Are we then to believe that the collective behavior we observe in fish is orthogonal to these additional reward signals? The authors do make mention of this in their discussion on limitations, but this is an ambiguity should in my opinion be addressed far earlier in the setup for the experiment. Again, the reason for this is because an argument of sufficiency alone is not exactly what I would consider solid ground for the claims the authors make about the advantage of their curiosity-driven, embodied-learning agents, since rule-based approaches have for some time given us equally sufficient learning rules that explain collective behavior equally well. This is not at all to say that I myself am not compelled by the argument of sufficiency, but if this alternative approach (which is often much more complicated than the rule-based approach) is to be taken seriously, it should likely go above and beyond in explaining certain phenomena (or providing more ecologically granular predictions) than the rule-based approach.

To summarize my first set of concerns, then, my sense is that neither the visual learning nor the reinforcement learning in this setup is challenging enough to empirically tout its advantage over a rule-based approach. The visual learning may effectively just be motion detection, and the reinforcement-learning requires nowhere near the multivariate complexity that ethologically realistic reinforcement-learning inherently entails. The argument the authors are making, then, reduces to an argument from first principles: Their approach is better because it uses raw sensory inputs and

embodiment. I WHOLEHEARTEDLY agree, but for this to be more than a philosophy paper, the empirical results should make it clear why this kind of learning does not ultimately reduce to the very same rule-learning the authors seem keen on challenging. I welcome any and all additional data or supplementary arguments the authors can muster to change my mind here. (I again also note that this comment is not meant to negate what I perceive to be the deep value of this paper; only to make it much stronger as a counterpoint to what already exists).

This is a great point. We agree with the Reviewer that this approach is valuable because it reduces to an argument from first principles by using raw sensory inputs and embodiment. We also agree that our paper could better emphasize why pixels-to-actions models have scientific value over and above rule-based models. In the revision, we focus on four arguments:

First, unlike rule-based models, pixels-to-actions models are well suited for exploring the **learning algorithms** that underlie collective behavior. Rule-based models do not generally learn, so they cannot address classic questions about the core learning algorithms that underlie social behavior. Since pixels-to-actions models learn from raw sensory inputs, and can be raised (trained) in the same visual environments as animals, these models can serve as **unified models** of the learning algorithms that drive social behavior in animals: whatever learning situation researchers give newborn animals can also be given to pixels-to-actions models. As shown in Fig. 1, this forms a closed-loop between the study of biological and artificial intelligence.

Second, since pixels-to-actions models can be directly compared with newborn animals in a closed-loop system, the models can generate new predictions for animal experiments (e.g., by first performing digital twin studies on artificial fish, then using their behavior as predictions of what biological fish will do in the same situations). This is also not something that rule-based models can do, since rule-based models do not ‘see’ what animals see and they do not produce actions in 3D visual environments.

Third, with pixels-to-actions models, we can determine the sufficient and necessary learning algorithms and experiences needed to produce animal behavior. For instance, the Reviewer asks: is collective behavior in fish orthogonal to additional reward signals? We provide an existence proof that it is, since additional rewards were not necessary to generate collective behavior. Our new experiment (Experiment 3) also shows classic social grouping effects (“us” vs. “them”) naturally emerge in these models, without needing to hardcode “social knowledge” into the brain. These social learning effects emerge when artificial fish are raised in visual environments with other agents (Fig. 5). As such, we provide scalable models that formalize both the learning algorithms and experiences that produce animal-like social abilities. Again, this is not something that can be done with rule-based models, since rule-based models do not learn from sensory experiences.

In this sense, we see these scientific inferences as analogous to those made in the study of core object recognition. Researchers have discovered that many details of biological visual systems (e.g., ion channels, spikes, rhythmic patterns, binocular vision) are not necessary for building machines that can recognize objects. Likewise, our results show that some reward signals (e.g., food, mates, predators) are not necessary to reproduce collective behavior; rather, generic learning algorithms (with no hardcoded knowledge of grouping) can drive embodied agents to group. By building artificial systems that learn in the same environments as animals, we can discover an engineering-level of abstraction for biological intelligence: a level close enough to biology to preserve the essential phenomena, but abstract enough to discard phenomena not required for reproducing biological intelligence. The question of which biological details matter for reproducing animal-like social behavior is thus turned into a testable question with pixels-to-actions models.

Fourth, we emphasize that pixels-to-actions models complement (not replace) rule-based models (pg. 7). Ultimately, a deep understanding of social behavior requires understanding at multiple levels, ranging from complex, neurally-mechanistic models that actually perform the same tasks as animals to simple, low-dimensional models that explain behavior using human-interpretable parameters. To this end, our study links complex neurally-mechanistic models (artificial fish) with simple rule-based models (attraction rules with a single parameter). This approach illuminates the algorithms driving social behavior, while simultaneously revealing simple attraction rules for understanding common developmental pathways and learning outcomes across biological and artificial systems.

Now, for my second major concern: Figures 2-3-4-5. It is exceedingly difficult to grok the main gist of the results from these figures while constantly moving my eyes back and forth from the biological animal reference to the artificial animal comparisons. Why not reprint the biological animal results on each of the subplots? It appears both the reference and the targets use the same scale in almost all comparisons (except in figure 5, but why not there as well?), so it seems pretty straightforward to combine them. For what it's worth, I think each result could actually be one single plot: with colors for the reference and targets, and linetypes (solid – dashed) for the actual data versus randomized control). It would also help to include a summary figure that uses some sort of fit statistic or distance metric to illustrate how far each model is from the reference. A nitpicky point, but many of the figures in their current form are also low-resolution, making it difficult to read axes and key bits of annotating text. These figures must in my opinion be improved prior to the manuscript being accepted.

This is a great critique. As suggested, we revised the figures so that the biological and artificial fish are shown on the same graphs (Fig. 2). This makes it much easier to see how they compare with one another across the metrics in the paper. We also improved the resolution of the figures by using vector-based figures.

In terms of minor comments:

The introductory literature review on the current state of DNN modeling for brains and behavior alike is somewhat oversimplistic and outdated. It is no longer the case, for example, that object recognition models are the dominant force in this kind of modeling – and in fact, most recent work in these domains almost exclusively leverages self-supervised (albeit disembodied) models. In terms of self-supervised embodied models in mice and fish, the works of Josh Merel and Bence Olvecky seem highly relevant here, but are not otherwise cited.

We appreciate the helpful critique. As suggested, we revised the introduction (pg. 2), emphasizing that researchers have made progress (1) building self-supervised learning algorithms, (2) training ANNs on biologically plausible data streams (e.g., head-mounted camera data from human infants), and (3) embodying ANNs to explore how artificial systems learn in more realistic learning contexts. We also cite the work of Josh Merel, Bence Olvecky, and many other researchers who have contributed to these advances.

Figure 5: The curve for the biological fish is the only truly exponential curve in this comparison; the rest (for the artificial fish) seem pretty sigmoidal. It does not appear to be the case that the authors meaningfully address this, but they probably should.

This is a great suggestion. While the biological fish were not reared for longer than 24 days, the authors did test adult fish, which had P s values (attraction rates) of 0.54. When this adult data are added to the graph of the biological fish, the pattern of development looks similar to the artificial fish (i.e., we see the “plateauing” behavior that the Reviewer predicts). To make this developmental commonality more clear, we revised the graph showing the biological and artificial fish (Fig. 2e).

REVIEWERS' COMMENTS

Reviewer #1 (Remarks to the Author):

The authors have done an excellent job with the revision and I now endorse the article for publication.

However, it would be nice to see three additional control experiments if the authors wish to pursue them. I would leave this up to the authors though. First, it would be interesting to see if schooling doesn't develop if there is another artificial stimulus in the visual scene that is more unpredictable. Relatedly, if the more-unpredictable stimulus is only introduced later, schooling shouldn't be disrupted. Lastly, it would be nice to show that blue+orange schools emerge if artificial fish are raised with both types.

Minor points:

1. In S3, I believe the y axis should be $p(0|0:1)$ not $p(1|1:2)$
2. The insets in 2f are all the same, as are the insets in 2g and S3. You only need the inset once per panel.
3. Typo: "What learning algorithms cause social preferences to develop so rapidly and flexibility early in life?" -> "What learning algorithms cause social preferences to develop so rapidly and flexibly early in life?"
4. Fig 4 caption: "(d) Results from the 2AFC talk." -> "(d) Results from the 2AFC task."

Reviewer #2 (Remarks to the Author):

I appreciate all the effort in addressing my concerns! The new experiment is very interesting. I think dropping the 2-eye part makes sense, and I appreciate your comments regarding plateauing behaviors.

Regarding experiment 3, I'd encourage the authors to give some more explanation as to why freezing the weights is the right experiment. Unless I'm misunderstanding the experiment, the fish don't see a stimulus at all like the novel fish before the test runs. From a statistical learning point of view, the null hypothesis should be that there's basically no coherent behavior to the novel stimulus, given the frozen network. This logic applies for all of the experiments in this section — in

each case, it's not surprising that the artificial fish don't develop preferences for stimuli that are unlike anything they had seen before.

I also think that the experiment (and the other experiments) could benefit from a stronger ablation — something simpler than an intrinsic reward signal, but still providing nontrivial rewards. It's not totally clear to me what the .001 curiosity strength ablation tells us. Isn't the curiosity reward the only reward the agent gets? In this case, the objective is not actually different (as it would be if that scaling were relative to some other term), but different behavior could be explained by the RL algorithm just not optimizing well at that scaling (and hence being close to random policy).

Not sure if this is something that's just an artifact of the review system, but I can't read all of the figures in the pdf, just in the original svg...some of the text is really small.

Reviewer #3 (Remarks to the Author):

My thanks to the authors for a thorough, comprehensive, and much improved revision. At a high-level, I believe almost all of my major concerns from the first review are satisfied by this revision. The added analyses (in particular, the addition of the more realistic environment and social preferences study) have contributed amply to my sense of confidence in the results. In short, I believe this revision could be accepted for publication effectively as is.

However, the one remaining major issue continues to be the figures – which, if anything – have become slightly more inscrutable to me this time around. Overwhelmingly, I think the issue is that they are far too busy. The authors could drastically improve the readability of their submission by moving the vast majority of the subfigures in each of their main figures to the appendix and adopting more of the “schematic” motif seen in their first figure.

To make this more concrete, I would at minimum like to see a figure that illustrates (with a single illustration per experiment) each of the major experiments in the analysis in a column on the left, and a single plot on the right, ideally with a single group-level contrast, a cross-bar plot, and some statistics. For example, the main result of experiment 1 could in my mind be summarized with the following question: “Did artificial fish reared using curiosity-driven learning algorithms develop biological fish-like grouping dynamics?” Plot is a categorical plot with two crossbars (mean + confidence intervals): Biological fish, artificial fish: Indication of chance accuracy; stats demonstrating greater than chance accuracy for the artificial fish. (Points can be individual models, if the authors so choose, but even this is extra information that is not entirely necessary to follow the main story.) Experiment 2: “Did this generalize to richer, more ecologically plausible environments?” Same biological fish bar; new artificial fish bar. All other information for each

experiment can be clarified in other plots – but at a glance, I think, it would be of major benefit to the paper if this “primary effect first”-style visualization was made more salient, front, and center.

While I do not think my lingering issues with the figures are enough to merit another full round of revisions, I HIGHLY encourage the authors to add some form of “gist figure” – and wherever possible, to limit the amount of information in any one plot. If all the information is relevant, I think the headings for each subfigure in a larger figure should be made more explicit – possibly even explicit questions that the particular subfigure should in theory answer. As it stands, each successive figure often feels like information overload.

On a more theoretical level, I do take slight issue still with one aspect of the argument made by the authors – again with respect to how we should “interpret” the sufficiency of general learning algorithms for reproducing “collective behavior”, and the absence of any “necessary” signal from ecologically relevant targets – food, mates, et cetera. If such learning rules are entirely sufficient for producing certain kinds of behavior like “grouping” – why don’t all organisms (which presumably could be learning from the same signals) behave the same? Obviously, one immediate answer here – which I’m sympathetic to – is that different organisms have different bodies – and that the combination of “specific bodies” and “general learning rules” is enough to mimic the suite of primary ecological pressures that produce ecologically relevant behaviors in any given organism. But the demonstration of this sufficiency then requires a “specificity” of body I’m not sure the authors have fully demonstrated here. Surely there are fish like the ones the authors have artificially imitated in this case that have all the same physiological effectors as zebrafish, but do not engage in the same kind of social grouping that zebrafish. What are we to make of the difference between these fish and zebrafish? This non-differentiability in the two-factor breakdown of “body” and “learning algorithm” is the reason I am somewhat suspicious of the absence of ecologically relevant targets – or in other words, why I think the authors might mention here that there are likely other cases in which (to mimic certain kinds of organism) you need not only “specificity of body” and “generality of learning algorithms – but also, “specificity of niche”. I should clarify that I offer this more as food for thought than an outright critique – but since the authors are pioneering this kind of science, I do think it might serve them to consider the emergent problem of “differentiating” species-specific behavior more carefully.

Issues with the figures aside and this more theoretical quibble aside, I again wish to applaud the authors on an excellent piece of work, and thank them for their careful, responsive revisions! Well done!

Response to Reviewers

Revised manuscript for *Nature Communications*
Parallel development of social behavior in biological and artificial fish
[NCOMMS-23-22656A]

Response to Reviewer #1

The authors have done an excellent job with the revision and I now endorse the article for publication.

However, it would be nice to see three additional control experiments if the authors wish to pursue them. I would leave this up to the authors though. First, it would be interesting to see if schooling doesn't develop if there is another artificial stimulus in the visual scene that is more unpredictable. Relatedly, if the more-unpredictable stimulus is only introduced later, schooling shouldn't be disrupted. Lastly, it would be nice to show that blue+orange schools emerge if artificial fish are raised with both types.

We appreciate the Reviewers' positive response to our revision. We also like their suggested control conditions. While we agree that these conditions may be interesting to run in the future, we do not currently have corresponding fish data for any of these conditions. Thus, it would not be possible to conclude whether or not the artificial fish learned like biological fish. Given that it is unclear what these extra control conditions would tell us (without corresponding fish data), we would prefer to leave these conditions for future studies, where researchers could extend our digital twin approach and collect parallel behavioral data from biological and artificial fish raised in these same environments.

Minor points:

1. In S3, I believe the y axis should be $p(0|0:1)$ not $p(1|1:2)$

Thank you for spotting this. The error has been corrected.

2. The insets in 2f are all the same, as are the insets in 2g and S3. You only need the inset once per panel.

As suggested, we have removed the repetitive insets from the figure.

3. Typo: "What learning algorithms cause social preferences to develop so rapidly and flexibility early in life?" -> "What learning algorithms cause social preferences to develop so rapidly and flexibly early in life?"

This typo has been corrected.

4. Fig 4 caption: "(d) Results from the 2AFC talk." -> "(d) Results from the 2AFC task."

This typo has been corrected.

Response to Reviewer #2

I appreciate all the effort in addressing my concerns! The new experiment is very interesting. I think dropping the 2-eye part makes sense, and I appreciate your comments regarding plateauing behaviors.

Regarding experiment 3, I'd encourage the authors to give some more explanation as to why freezing the weights is the right experiment. Unless I'm misunderstanding the experiment, the fish don't see a stimulus at all like the novel fish before the test runs. From a statistical learning point of view, the null hypothesis should be that there's basically no coherent behavior to the novel stimulus, given the frozen network. This logic applies for all of the experiments in this section — in each case, it's not surprising that the artificial fish don't develop preferences for stimuli that are unlike anything they had seen before.

This is a good suggestion. We froze the weight in the artificial fish to mimic the sensitive/critical period of early visual learning. In most precocial animals (including fish), there are sensitive and critical periods which slow and sometimes stop learning. This means that much of an animal's behavior will be based on experiences that happened early in life, rather than recently. Conversely, ANNs do not have sensitive/critical periods, so an embodied ANN's behavior will largely be based on their recent experiences. This poses a problem for comparing animals to ANNs. One solution is to gradually slow the learning rate of ANNs across the training phase and to then freeze the network weights when the learning rate hits zero. We see this as roughly analogous to critical periods in brains, in which learning can gradually slow down and then halt in young animals.

To clarify the importance of critical periods to the reader, our Discussion contains the following text (pg. 6-7):

We hypothesize that curiosity produces social behavior because social partners are the least predictable things in a newborn's visual environment. Curiosity-driven learning systems are attracted to unpredictable things and will learn to produce actions that lead to unpredictable outcomes (e.g., actions that keep social agents in view). Curiosity-driven systems should thus develop social behavior when (1) they are raised/trained in environments with social partners (e.g., parents and siblings) and (2) there is a critical period (i.e., cessation of learning), which are widespread in animals, especially during early brain development⁷⁶⁻⁷⁹. Under these conditions, we have shown that generic learning systems can rapidly develop core social skills.

In Experiment 3, we also added a brief justification (pg. 5):

After training, we froze the ANN weights to mimic sensitive/critical periods observed in animals. Many animals have sensitive/critical periods which slow/stop learning, meaning part of an animal's behavior can be based on experiences that happened early in life, rather than recently. ANNs do not have sensitive/critical periods, which poses a problem for comparing animals to ANNs. One solution is to gradually slow the learning rate of ANNs across the training phase and freeze the weights when learning hits zero. We see this as roughly analogous to critical periods in brains, where learning can gradually slow and cease in animals.

I also think that the experiment (and the other experiments) could benefit from a stronger ablation — something simpler than an intrinsic reward signal, but still providing nontrivial rewards. It's not totally clear to me what the .001 curiosity strength ablation tells us. Isn't the curiosity reward the only reward the agent gets? In this case, the objective is not actually different (as it would be if that scaling were relative to some other term), but different behavior could be explained by the RL algorithm just not optimizing well at that scaling (and hence being close to random policy).

In our prior revision, we did add one strong ablation (removing social experiences from the training data, Fig. 6). In this situation, the artificial fish no longer learned social preferences. We also think the curiosity reduction ablation (reducing the curiosity reward) is informative because it shows that the curiosity reward is important for learning social behavior. It could have been the case, for example, that all artificial fish trained in social environments show this pattern, regardless of the strength of

their intrinsic reward. Together, these two ablations show that (1) social experiences and (2) strong intrinsic rewards are essential ingredients for learning social preferences.

Not sure if this is something that's just an artifact of the review system, but I can't read all of the figures in the pdf, just in the original svg...some of the text is really small.

We believe that this is an artifact of the reviewer system. However, we will ensure that all figures have readable text (Reviewer #3 also suggested we alter the figures to make them more clear).

Response to Reviewer #3

My thanks to the authors for a thorough, comprehensive, and much improved revision. At a high-level, I believe almost all of my major concerns from the first review are satisfied by this revision. The added analyses (in particular, the addition of the more realistic environment and social preferences study) have contributed amply to my sense of confidence in the results. In short, I believe this revision could be accepted for publication effectively as is.

However, the one remaining major issue continues to be the figures – which, if anything – have become slightly more inscrutable to me this time around. Overwhelmingly, I think the issue is that they are far too busy. The authors could drastically improve the readability of their submission by moving the vast majority of the subfigures in each of their main figures to the appendix and adopting more of the “schematic” motif seen in their first figure.

We appreciate the critique of our figures. In this revision, we have substantially revised the figures to make them less busy and more clear in terms of what the data represents. As suggested, for each major analysis and experiment, we now include an illustration showing what the measure means and how the data were calculated. We also now avoid having more than 1-2 analyses in a single figure, which helps keep the figures from being too crowded.

To make this more concrete, I would at minimum like to see a figure that illustrates (with a single illustration per experiment) each of the major experiments in the analysis in a column on the left, and a single plot on the right, ideally with a single group-level contrast, a cross-bar plot, and some statistics. For example, the main result of experiment 1 could in my mind be summarized with the following question: “Did artificial fish reared using curiosity-driven learning algorithms develop biological fish-like grouping dynamics?” Plot is a categorical plot with two crossbars (mean + confidence intervals): Biological fish, artificial fish: Indication of chance accuracy; stats demonstrating greater than chance accuracy for the artificial fish. (Points can be individual models, if the authors so choose, but even this is extra information that is not entirely necessary to follow the main story.) Experiment 2: “Did this generalize to richer, more ecologically plausible environments?” Same biological fish bar; new artificial fish bar. All other information for each experiment can be clarified in other plots – but at a glance, I think, it would be of major benefit to the paper if this “primary effect first”-style visualization was made more salient, front, and center.

As suggested by the Reviewer and in the Editor's checklist, we added a summary Figure (Fig. 8) that communicates the main result/advance as the final figure.

While I do not think my lingering issues with the figures are enough to merit another full round of revisions, I HIGHLY encourage the authors to add some form of “gist figure” – and wherever possible, to limit the amount of information in any one plot. If all the information is relevant, I think the headings for each subfigure in a larger figure should be made more explicit – possibly even explicit questions that the particular subfigure should in theory answer. As it stands, each successive figure often feels like information overload.

As discussed above, we significantly modified the figures to make them more digestible. As suggested by the Reviewer, we also (1) added headings describing the specific question under investigation in the figure and (2) illustrations showing what we measured in each figure. We believe the new figures are more simple and easy to understand, and (hopefully) no longer overload the reader.

On a more theoretical level, I do take slight issue still with one aspect of the argument made by the authors – again with respect to how we should “interpret” the sufficiency of general learning algorithms for reproducing “collective behavior”, and the absence of any “necessary” signal from ecologically relevant targets – food, mates, et cetera. If such learning rules are entirely sufficient for producing certain kinds of behavior like “grouping” – why don’t all organisms (which presumably could be learning from the same signals) behave the same? Obviously, one immediate answer here – which I’m sympathetic to – is that different organisms have different bodies – and that the combination of “specific bodies” and “general learning rules” is enough to mimic the suite of primary ecological pressures that produce ecologically relevant behaviors in any given organism. But the demonstration of this sufficiency then requires a “specificity” of body I’m not sure the authors have fully demonstrated here. Surely there are fish like the ones the authors have artificially imitated in this case that have all the same physiological effectors as zebrafish, but do not engage in the same kind of social grouping that zebrafish. What are we to make of the difference between these fish and zebrafish? This non-differentiability in the two-factor breakdown of “body” and “learning algorithm” is the reason I am somewhat suspicious of the absence of ecologically relevant targets – or in other words, why I think the authors might mention here that there are likely other cases in which (to mimic certain kinds of organism) you need not only “specificity of body” and “generality of learning algorithms – but also, “specificity of niche”. I should clarify that I offer this more as food for thought than an outright critique – but since the authors are pioneering this kind of science, I do think it might serve them to consider the emergent problem of “differentiating” species-specific behavior more carefully.

This is an excellent critique. The question of what causes species specific behavior is a core question underlying our work, and we agree with the Reviewer that we could be more specific.

Given the importance of social experiences for learning social behavior in our models, we predict that social animals will have richer social experiences than solitary animals. Support for this prediction comes from observations that solitary animals (e.g., tigers, leopards, bears, wolverines, orangutans) tend to have just 1-3 offspring per litter, whereas social animals often have litters with dozens of offspring (rabbits), produce hundreds of fry (guppies, zebrafish), or lay thousands of eggs (ants, bees). We also predict that animals who typically have one offspring at a time but live in rich social worlds (e.g., humans) will still learn robust social behavior because they acquire rich social experiences early in life.

We also agree that our paper could use a more detailed explanation of what we mean by ‘body’ and how body differences could help fit animal species to different niches. To do so, we added the following text to the paper (pg. 8):

Importantly, the body of an animal includes not only its general morphology, but also its physiology (e.g., metabolism, thermoregulation, reproductive mechanisms) and sensors (e.g., eyes, ears, nose). If animals have different sensors, for example, then their brains will receive different information. Behavioral differences across animals could thus be due to differences in brains, bodies, or environments. All of these factors influence how animals fit to different evolutionary niches.

Issues with the figures aside and this more theoretical quibble aside, I again wish to applaud the authors on an excellent piece of work, and thank them for their careful, responsive revisions! Well done!

Thank you!